# Near-Optimal Sample Complexity for Online Constrained MDPs

**Chang Liu**
University of California, Los Angeles
changliu11@ucla.edu

**Yunfan Li**
University of California, Los Angeles
yunfanli@ucla.edu

**Lin F. Yang**
University of California, Los Angeles
linyang@ee.ucla.edu

## Abstract

Safety is a fundamental challenge in reinforcement learning (RL), particularly in real-world applications such as autonomous driving, robotics, and healthcare. To address this, Constrained Markov Decision Processes (CMDPs) are commonly used to enforce safety constraints while optimizing performance. However, existing methods often suffer from significant safety violations or require a high sample complexity to generate near-optimal policies. We address two settings: relaxed feasibility, where small violations are allowed, and strict feasibility, where no violation is allowed. We propose a model-based primal-dual algorithm that balances regret and bounded constraint violations, drawing on techniques from online RL and constrained optimization. For relaxed feasibility, we prove that our algorithm returns an $\varepsilon$-optimal policy with $\varepsilon$-bounded violation with arbitrarily high probability, requiring $\widetilde{O}\left(\frac{SAH^3}{\varepsilon^2}\right)$ learning episodes, matching the lower bound for unconstrained MDPs. For strict feasibility, we prove that our algorithm returns an $\varepsilon$-optimal policy with zero violation with arbitrarily high probability, requiring $\widetilde{O}\left(\frac{SAH^5}{\varepsilon^2\zeta^2}\right)$ learning episodes, where $\zeta$ is the problem-dependent Slater constant characterizing the size of the feasible region. This result matches the lower bound for learning CMDPs with access to a generative model. Our results demonstrate that learning CMDPs in an online setting is as easy as learning with a generative model and is no more challenging than learning unconstrained MDPs when small violations are allowed.

## 1 Introduction

Safety is a fundamental concern in reinforcement learning (RL), especially in real-world applications such as autonomous driving [1], healthcare [2], and industrial automation [3]. Constrained Markov Decision Processes (CMDPs) [4] are widely used to ensure safety by incorporating safe constraints and safe policies into the decision-making process. These frameworks allow for optimizing performance while limiting risky actions in safety-critical environments, such as preventing collisions in autonomous vehicles or ensuring correct treatment in healthcare. However, during the course of training, many RL methods can suffer from significant safety violations [5, 6], particularly when exploring new states or actions. This is highly problematic in real-world systems where even temporary unsafe actions can result in accidents, equipment damage, or hazardous conditions. For instance, in autonomous driving [7], safety violations during training might lead to collisions or dangerous maneuvers, while in robotics [8], such violations could cause physical harm to equipment

39th Conference on Neural Information Processing Systems (NeurIPS 2025).

or workers. As a result, it is crucial to design methods that guarantee bounded safety constraint violations, ensuring that any violations during training remain within acceptable limits, thereby preventing catastrophic outcomes while maintaining safety throughout the learning process.

Tabular episodic MDPs, one of the most fundamental settings in RL, have been extensively studied, with various methods proposed to solve them [9, 10, 11, 12, 13, 14, 15]. Many algorithms have been shown to achieve minimax optimal regret $\widetilde{O}(\sqrt{SAH^3K})$ and [15] provided a method to achieve the minimax optimal sample complexity $\widetilde{O}\left(\frac{SAH^3}{\varepsilon^2}\right)$. Extending to CMDPs, prior works [6, 16, 17, 18, 19, 20, 21, 22, 23, 24] have proposed methods in various settings that either achieve sublinear regret or return near-optimal algorithms. However, lower bounds and algorithms with provable minimax optimal upper bounds still remain unexplored in most of these works. Two primary feasibility settings in CMDPs are **relaxed feasibility** and **strict feasibility** [18, 20, 21, 23]. In the relaxed feasibility setting, algorithms may return approximately optimal policies that allow small constraint violations. In contrast, in the strict feasibility setting, the learned policies must be near-optimal while ensuring zero constraint violations. The formal formulations for both settings are defined in section 3. [18] proposed a model-based algorithm that achieves minimax sample complexity in both regimes, but relied on the assumption of a generative model that grants the learner random access to any state-action queries. However, in many real-world scenarios such as robotics and autonomous driving, the learning agent has to collect data online through sequential interactions with the environment. How or whether a minimax optimal sample complexity can be achieved in the online setting, which generalizes the random access setting and is inherently more challenging [25], remains unknown. This raises the following question:

*Can we design safe **online** reinforcement learning (RL) algorithms for both **relaxed and strict feasibility** settings that achieve **near-optimal sample efficiency** in large-scale state spaces and long horizons while guaranteeing **approximate reward optimality** with arbitrarily high probability?*

In this paper, we affirmatively answer this question by proposing a model-based primal-dual algorithm. Our contributions are summarized as follows:

- We propose a model-based primal-dual algorithm with doubling batch updates. In each episode, a policy is derived by mixing policies from multiple iterations of primal-dual updates in a constrained model represented in its Lagrangian form. This policy is then used to gather data to update the empirical transition model. The doubling batch update technique ensures a lazy update of the empirical transition model, allowing us to derive tighter theoretical bounds.

- For the relaxed feasibility setting, we prove that our algorithm returns an $\varepsilon$-optimal policy with at most $\varepsilon$ constraint violation after $\widetilde{O}\left(\frac{SAH^3}{\varepsilon^2}\right)$ of online learning episodes, where $S$ and $A$ is the number of states and actions, $H$ is the horizon. This sample complexity matches the lower bound for unconstrained MDPs: $\Omega\left(\frac{SAH^3}{\varepsilon^2}\right)$ [11], showing that learning CMDPs in the relaxed feasibility setting is as easy as learning unconstrained MDPs.

- For the strict feasibility setting, we prove that our algorithm returns an $\varepsilon$-optimal zero-violation policy after $\widetilde{O}\left(\frac{SAH^5}{\varepsilon^2\zeta^2}\right)$ of online learning episodes, where $\zeta$ is the problem-dependent Slater constant characterizing the size of the feasible region. This sample complexity matches the lower bound for infinite-horizon discounted CMDPs with access to a generative model: $\Omega\left(\frac{SA}{(1-\gamma)^5\varepsilon^2\zeta^2}\right)$ [18], showing that learning CMDPs with online access is as easy as learning CMDPs with random access.

**Notation.** We introduce a set of notation to be used throughout. For any two vectors $x, y \in \mathbb{R}^d$ with the same dimension $d$, we use $xy$ to abbreviate the inner product $x^\top y$, e.g. $P_{s,a,h}V^*_{h+1,r} = \sum_{s'} P_{s,a,h}(s')V^*_{h+1,r}(s')$. For any integer $S > 0$, any probability vector $p \in \Delta_{[S]}$ and another vector $v = [v_i]_{1 \leq i \leq S}$, we denote by $\mathbb{V}(p, v) := \langle p, v^2 \rangle - (\langle p, v \rangle)^2$ the associated variance, where $v^2 = [v_i^2]_{1 \leq i \leq S}$ represents element-wise square of $v$.

## 2 Related Work

**Constrained Markov Decision Process (CMDP)**   The Constrained Markov Decision Process (CMDP) [4] is a key model for addressing safety concerns in reinforcement learning (RL). Many existing works on CMDPs employ a primal-dual approach to achieve sublinear regret while maintaining bounded constraint violations [18, 26, 27, 28, 29, 30]. Another widely-used method is adapting policy gradient algorithms [31, 32, 33, 34]. Furthermore, [6] introduces a more stringent metric for hard constraint violation, where only positive constraint violations are accumulated. Their approach achieves sublinear regret, constraint violations and hard constraint violation. Recently, [35] extended this idea to a linear setting, obtaining similar results. In practical applications, ensuring strict adherence to safety constraints without violations often requires system-specific assumptions. For instance, [36] assumes regularity in the safety functions, while [37] presumes knowledge of a safe action for each state. Additionally, [16, 17] assume the existence of a known safe policy and its true constraint value, achieving improved regret bounds and constraint violations compared to [6]. Building on the assumption of a known safe policy and its true constraint value, our work proposes a primal-dual low-switching algorithm, leveraging advanced techniques from standard MDPs. This approach not only improves the regret bound but also maintains a constant constraint violation. [18] solved the infinite-horizon discounted tabular CMDPs in the random access setting. Our work extends their results to tabular CMDPs in the online setting, which serve as a crucial benchmark for safe RL that provides a structured environment for theoretical analysis and algorithmic validation. Further extension to function approximation or multiple constraints is out of the scope of this paper and an interesting topic for future research.

**Episodic unconstrained MDPs**   [38] first provided an upper bound of $O(\sqrt{S^2AKHD^2})$. [39, 11] established lower bounds of $\Omega(\sqrt{SAH^3K})$ for regret and $\Omega(SAH^3/\varepsilon^2)$ for sample complexity. [40] introduced a posterior sampling approach for RL, achieving minimax-optimal regret bounds of $O(\sqrt{SAHK})$ under certain conditions. [9] further achieved a minimax optimal regret for a model-based algorithm, and [41] developed similar results for a model-free Q-learning method. Many methods have achieved near-optimal upper bounds [12, 13, 14]. Recently, [15] achieved minimax optimal regret, sample complexity, and burn-in cost, yet such methods are rarely explored in CMDPs. To our knowledge, this work is the first to incorporate state-of-the-art techniques for unconstrained MDPs into CMDPs, achieving minimax optimal performance in online constrained RL.

## 3 Problem Setup

We consider a finite-horizon non-stationary constrained Markov Decision Process (MDP) defined by the tuple $M = (\mathcal{S}, \mathcal{A}, H, P, r, c, b)$, where $\mathcal{S}$ is the state space, $\mathcal{A}$ is the action space, and $H$ is the horizon length. The unknown transition probability at each time step is denoted by $P_{s,a,h}$, where $P_{s,a,h}(s')$ represents the probability of transitioning to state $s'$ from state $s$ after taking action $a$ at time step $h$. The reward function $r_h : \mathcal{S} \times \mathcal{A} \rightarrow [0,1]$ quantifies the immediate reward the agent receives for taking action $a$ in state $s$ at time step $h$. Similarly, the cost function $c_h : \mathcal{S} \times \mathcal{A} \rightarrow [0,1]$ represents safety violations incurred for the same action. We assume that both the reward and cost functions are known to the agent, though the same results can be easily extended to the case where neither function is known. Finally, $b \in (0, H]$ is a predefined safety constraint that limits the cumulative cost over the episode.

The agent interacts with the environment over $K$ episodes, each consisting of $H$ steps. At the start of each episode $k$, the agent selects a randomized policy $\pi^k = \{\pi_h^k\}_h$, where at time step $h$ the policy $\pi_h^k : \mathcal{S} \rightarrow \Delta_{\mathcal{A}}$ prescribes a distribution over actions conditioned on the current state. The policy is executed with the goal of maximizing the cumulative reward while ensuring that the cumulative cost remains within the safety limit. The cumulative value at state $s$ and time step $h$, with respect to any function $g : \mathcal{S} \times \mathcal{A} \rightarrow \mathbb{R}$, under policy $\pi$, is defined as: $V_{h,g}^\pi(s) = \mathbb{E}_{P,\pi}\left[\sum_{t=h}^H g(S_t, A_t)\Big|S_h = s\right]$, representing the expected cumulative sum of $g(S_t, A_t)$ from time step $h$ to the end of the episode, given that the process starts in state $s$ at time $h$. The objective of CMDP is to solve the following constrained optimization problem:

$$\max_\pi V_{1,r}^\pi(s_1) \quad \text{s.t.} \quad V_{1,c}^\pi(s_1) \leq b, \tag{1}$$

where $V_{1,r}^{\pi}(s_1)$ is the expected cumulative reward value, and $V_{1,c}^{\pi}(s_1)$ is the expected cumulative cost value, constrained by the safety threshold $b$. The optimal policy $\pi^*$ solving eq. (1) is defined as

$$\pi^* = \arg\max_{\pi} V_{1,r}^{\pi}(s_1) \quad \text{s.t.} \quad V_{1,c}^{\pi}(s_1) \le b.$$

The corresponding expected reward and cost value are denoted by $V_{1,r}^*(s_1)$ and $V_{1,c}^*(s_1)$. For a CMDP instance, we define the Slater constant $\zeta := \max_{\pi} b - V_{1,c}^{\pi}(s_1)$. The Slater constant $\zeta$ measures the size of the feasible region. We also define the policy that with the minimal cost value: $\pi_c^* \in \arg\max_{\pi} b - V_{1,c}^{\pi}(s_1)$.

We now define the regret and constraint violation over $K$ episodes:

$$\text{Regret}(K) := \sum_{k=1}^{K} \left( V_{1,r}^*(s_1^k) - V_{1,r}^{\pi^k}(s_1^k) \right) \quad \text{and} \quad \text{CV}(K) := \left( \sum_{k=1}^{K} \left( V_{1,c}^{\pi^k}(s_1^k) - b \right) \right)_+ . \quad (2)$$

We will provide upper bounds on both regret and constraint violation of our algorithm, but more importantly, convert such bounds to our final sample complexity bounds. For a small suboptimality error $\varepsilon$, we define the two settings as follows:

**Relaxed feasibility:** The agent's objective is to return a policy that with high probability achieves an approximately optimal reward value, while violates the constraint by a small margin. Formally the agent returns a policy $\pi$ such that with high probability,

$$V_{1,r}^{\pi}(s_1) \ge V_{1,r}^*(s_1) - \varepsilon, \quad \text{and} \quad V_{1,c}^{\pi}(s_1) \le b + \varepsilon. \quad (3)$$

**Strict feasibility:** The agent's objective is to return a policy that with high probability achieves an approximately optimal reward value, and simultaneously achieves zero constraint violation. Formally the agent returns a policy $\pi$ such that with high probability,

$$V_{1,r}^{\pi}(s_1) \ge V_{1,r}^*(s_1) - \varepsilon, \quad \text{and} \quad V_{1,c}^{\pi}(s_1) \le b. \quad (4)$$

In the next section, we present our technical methodology to tackle both settings in online CMDPs.

## 4 Methodology

Before introducing our method, we examine the state-of-the-art algorithms in tabular online CMDPs, and discuss why their methods fail to achieve near-optimal sample complexity.

One of the biggest challenges in analyzing model-based algorithms in unconstrained MDPs and CMDPs is to decouple the correlation between the empirical model $\widehat{P}$ and the estimate values $\widehat{V}^{\pi^k}$. Indeed, both [16] and [17] get the extra regret terms due to loose decoupling steps. In particular, [17] achieve a regret of $\widetilde{O}(\sqrt{S^2 A H^6 K}/(b - c^0))$, where $c^0$ is the value of a known safe policy. In both their regret analysis, the authors bound the term $\widehat{V}_{1,r}^{\pi^k}(s_1) - V_{1,r}^{\pi^k}(s_1)$ by

$$\widehat{V}_{1,r}^{\pi^k}(s_1) - V_{1,r}^{\pi^k}(s_1) = \sum_{h=1}^{H} \mathbb{E}_{\pi^k, P} \left[ \sum_{s'} \left( \widehat{P}_{s_h^k, a_h^k, h}(s') - P_{s_h^k, a_h^k, h}(s') \right) \widehat{V}_{h+1, r}^{\pi^k}(s') \right]$$

$$\le \sum_{h=1}^{H} \mathbb{E}_{\pi^k, P} \left[ \sum_{s'} \left| \widehat{P}_{s_h^k, a_h^k, h}(s') - P_{s_h^k, a_h^k, h}(s') \right| H \right],$$

where the equality follows from value difference lemma and the inequality follows from Hölder's inequality and $\widehat{V} \le H$. Then they invoke Bernstein's inequality on the concentration bound of $\widehat{P}$. This technique is also known to be used in unconstrained MDPs [42] and gives a regret of $\widetilde{O}(\sqrt{S^2 A H^4 K})$, which is suboptimal compared to the lower bound $\Omega(\sqrt{S A H^3 K})$. Following [42], subsequent works [9, 15] introduced different techniques to achieve near-optimal regret, but neither method extends directly to CMDPs.

In the regret analysis of [9], the authors bound $\widehat{V}_{h, \widetilde{r}}^{\pi^k}(s_h^k) - V_{h, r}^{\pi^k}(s_h^k)$ by writing it in a recursive form. One of the key steps is to bound the following term as:

$$\left( \widehat{P}_{s, a, h} - P_{s, a, h} \right) \left( \widehat{V}_{h+1, \widetilde{r}}^{\pi^k} - V_{h+1, r}^* \right) \le \sum_{s'} \left| \widehat{P}_{s, a, h}(s') - P_{s, a, h}(s') \right| \cdot \left( \widehat{V}_{h+1, \widetilde{r}}^{\pi^k}(s') - V_{h+1, r}^{\pi^k}(s') \right),$$

where the inequality follows from the optimism of $\widehat{V}_{h,\widetilde{r}}^{\pi^k}$ and optimality of $V_{h,r}^*$: $\widehat{V}_{h,\widetilde{r}}^{\pi^k}(s) \geq V_{h,r}^*(s) \geq V_{h,r}^{\pi^k}(s), \forall h \in [H], k \in [K], s \in \mathcal{S}$. This argument fails in CMDPs, because the optimal policy $\pi^*$ in CMDPs is not necessarily optimal for all $h \in [H]$.

The high-level idea of [15] to decouple the correlation between $\widehat{P}$ and $\widehat{V}$ is that, when fixing a "profile" that keeps track of visitation counts for all $(s, a, h)$ tuples, $\widehat{V}_{h+1}$ is independent of $\widehat{P}_h$. A union bound over all possible profiles then gives the desired regret bound. To avoid the number of such profiles being exponential in $K$, the authors adopt the double batch updates explained later. Although decoupling arises naturally in unconstrained MDPs through backward updates, it is not the case in constrained MDPs. If the policy $\pi$ is determined by enforcing the constraint $\widehat{V}_{1,c}^{\pi} \leq b$, then $\pi$ and thus $\widehat{V}_{h+1}^{\pi}$ will be inherently correlated to all $\widehat{P}_h$'s.

To tackle these challenges, we use a model-based approach to address the CMDP problem defined in eq. (1). Adopting the doubling batch updates, we update our empirical transition matrix only when the visitation count of any state-action pair doubles. To be specific, we denote $\bar{N}_h(s, a)$ as the total visitation count of state-action pair $(s, a)$ in time step $h$, $N_h(s, a, s')$ as the count of transitions from $(s, a)$ to $s'$ since the last update, and $N_h(s, a) = \sum_{s'} N_h(s, a, s')$ as the visitation count of $(s, a)$ since the last update. We update an empirical transition matrix $\widehat{P}$ whenever $\bar{N}_h(s, a)$ for any $(s, a)$ doubles, such that $\widehat{P}_{s,a,h}(s') = \frac{N_h(s,a,s')}{N_h(s,a)}$. Note that we will only use the data collected after the last update to calculate $\widehat{P}$. With $\widehat{P}$, we are able to formulate an empirical CMDP.

We adopt a UCB-style bonus for both reward and cost. For any reward function $g$ and policy $\pi$, we define the bonus for a $(s, a, h, k)$ tuple as

$$b_{h,g}^{k,\pi}(s, a) = c_1 \sqrt{\frac{\mathbb{V}(\widehat{P}_{s,a,h}, \widehat{V}_{h+1,g}^{\pi}) \log(1/\delta')}{N_h(s, a)}} + c_2 \frac{H \log(1/\delta')}{N_h(s, a)}, \tag{5}$$

where $c_1$ and $c_2$ are constant to be specified later and $\delta' = \delta/(200SAH^2K^2)$ is related to the confidence level $\delta$. For reward, we add this Bernstein-style bonus $b_{h,r}(s, a)$ to $r_h(s, a)$ for each $(s, a)$ to encourage exploration. We denote the optimistically biased reward estimate as $\widetilde{r}$, i.e., $\widetilde{r}_h(s, a) = r_h(s, a) + b_{h,r}(s, a)$. For safety cost, we subtract a Bernstein-style bonus $b_{h,c}(s, a)$ from $c_h(s, a)$. We denote the optimistically biased cost estimate by $\underline{c}$, i.e., $\underline{c}_h(s, a) = c_h(s, a) - b_{h,c}(s, a)$. By using the optimistically biased cost estimate we will underestimate the cumulative cost. To compensate for this and strive to satisfy the safety constraint in the strict feasibility setting, we define a pessimistic constraint constant $b'$ for each episode by subtracting a gap $\Delta$ from $b$, that is, $b' := b - \Delta$. Whereas for the relaxed feasibility setting, since small violations are allowed, we define an optimistic constraint constant $b' := b + \tau$. This non-negative optimistic shift $\tau$ is also required to derive $\zeta$-free results for the relaxed feasibility setting. We will specify the value of $\Delta$ and $\tau$ later.

We now introduce an empirical CMDP for each episode $k$, defined by $\widehat{M}_k = (\mathcal{S}, \mathcal{A}, H, \widehat{P}, \widetilde{r}, \underline{c}, b')$, and the corresponding optimization problem $\widehat{\mathcal{P}}^k$:

$$\max_{\pi} \widehat{V}_{1,\widetilde{r}}^{\pi}(s_1) \quad \text{s.t.} \quad \widehat{V}_{1,\underline{c}}^{\pi}(s_1) \leq b'. \tag{6}$$

As discussed before, directly solving eq. (6) results in complex correlation of $\widehat{P}$ and $\widehat{V}^{\pi}$, leading to suboptimal results. Therefore, we employ a primal-dual approach, which transforms the constrained optimization problem into a saddle-point problem. Let $\lambda \geq 0$ be the dual variable associated with the cost constraint. The equivalent saddle-point problem to eq. (6) is:

$$\min_{\lambda \geq 0} \max_{\pi} \widehat{V}_{1,\widetilde{r}}^{\pi}(s_1) - \lambda \left( \widehat{V}_{1,\underline{c}}^{\pi}(s_1) - b' \right), \tag{7}$$

and we denote $(\widehat{\pi}^{k,*}, \widehat{\lambda}^{k,*})$ as the optimal solutions to the saddle point problem eq. (7).

We solve the saddle-point problem eq. (7) iteratively, and for each iteration $t \in [T]$, we alternatively update iterates of the primal variable $\widehat{\pi}_t^k$ and the dual variable $\widehat{\lambda}_{t+1}^k$. The primal update is

$$\widehat{\pi}_t^k = \arg\max_{\pi} \widehat{V}_{1,\widetilde{r}}^{\pi}(s_1) - \widehat{\lambda}_t^k \left( \widehat{V}_{1,\underline{c}}^{\pi}(s_1) - b' \right) = \arg\max_{\pi} \widehat{V}_{1,\widetilde{r}-\widehat{\lambda}_t^k \underline{c}}^{\pi}(s_1), \tag{8}$$

where the last equality follows from lemma 15. By augmenting rewards with the Lagrange-weighted costs, the policy $\widehat{\pi}_t^k$ can be solved with backward updates, allowing us to decouple $\widehat{V}_{h+1}^\pi$ from $\widehat{P}_h$. However, the value estimates $\widehat{V}^\pi$ now depend on the dual variable $\widehat{\lambda}^k$ that takes continuous values. To retain tractable union bounds in the analysis, we must restrict $\widehat{\lambda}$ to a discrete set. Specifically, we update the dual variable by performing a single gradient descent step with step size $\eta$. We then discretize the values by rounding the gradient descent result to the nearest element in an $\varepsilon$-net $\Lambda = \{0, \varepsilon_1, 2\varepsilon_1, \ldots, U\}$. The resulting dual update is

$$\widehat{\lambda}_{t+1}^k = \mathcal{R}_\Lambda \left[ \widehat{\lambda}_t^k + \eta \left( \widehat{V}_{1,\underline{c}}^{\widehat{\pi}_t^k}(s_1) - b' \right) \right], \tag{9}$$

where $\mathcal{R}_\Lambda(\lambda) = \arg\min_{p \in \Lambda} |p - \lambda|$ is a rounding function.

---

**Algorithm 1:** Model-based algorithm for online CMDP

---

**Input:** $\mathcal{S}, \mathcal{A}, H, K, r, c, \zeta, b', c_1 = 460/9, c_2 = 544/9, \eta = \frac{U}{H\sqrt{T}}, T, \varepsilon_1, U$.

**Initialization:** for all $k \in [K]$, $\widehat{\lambda}_1^k \leftarrow 0$, for all $(s, a, s', h)$, set $N_h(s, a, s') \leftarrow 0$,
$\qquad\qquad \bar{N}_h(s, a, s') \leftarrow 0, N_h(s, a) \leftarrow 0$; for all $\pi$, set $\widehat{V}_{h,\underline{c}}^\pi(s) \leftarrow 0, \widehat{V}_{h,\widetilde{r}}^\pi(s) \leftarrow H$.

**for** $k = 1, \cdots, K$ **do**

    **for** $t = 1, \cdots, T$ **do**

        $\widehat{\pi}_t^k = \arg\max_\pi \widehat{V}_{1,\widetilde{r}}^\pi(s_1^k) - \widehat{\lambda}_t^k \widehat{V}_{1,\underline{c}}^\pi(s_1^k)$

        $\widehat{\lambda}_{t+1}^k = \mathcal{R}_\Lambda[\widehat{\lambda}_t^k - \eta(b' - \widehat{V}_{1,\underline{c}}^{\widehat{\pi}_t^k}(s_1^k))]$

    $\pi^k = \frac{1}{T} \sum_{t=1}^T \widehat{\pi}_t^k$

    **for** $h = 1, \cdots, H$ **do**

        Observe $s_h^k$, take action $a_h^k \sim \pi_h^k(\cdot|s_h^k)$, receive $r_h^k, c_h^k$, observe $s_{h+1}^k$

        $(s, a, s') \leftarrow s_h^k, a_h^k, s_{h+1}^k$

        $\bar{N}_h(s, a) \leftarrow \bar{N}_h(s, a) + 1, N_h(s, a, s') \leftarrow N_h(s, a, s') + 1$

        **if** $\bar{N}_h(s, a) \in \{1, 2, 4, \cdots, 2^{\log_2 K}\}$ **then**

            $N_h(s, a) \leftarrow \sum_{\widetilde{s}} N_h(s, a, \widetilde{s})$

            $\widehat{P}_{s,a,h}(\widetilde{s}) \leftarrow N_h(s, a, \widetilde{s})/N_h(s, a)$

            TRIGGERED $\leftarrow$ TRUE

            $N_h(s, a, \cdot) \leftarrow 0$

    **if** *TRIGGERED* **then**

        TRIGGERED $\leftarrow$ FALSE

        $\widehat{V}_{H+1,g}^\pi(s) \leftarrow 0, \forall x \in \mathcal{S}$

        **for** $h = H, H-1, \cdots, 1$ **do**

            **for** $(s, a) \in \mathcal{S} \times \mathcal{A}$ *and any* $\pi$ **do**

                $\widehat{Q}_{h,\widetilde{r}}^\pi(s, a) = \min\{r_h(s, a) + b_{h,r}^{k,t,\pi}(s, a) + \widehat{P}_{s,a,h}\widehat{V}_{h+1,\widetilde{r}}^\pi, H\}$

                $\widehat{V}_{h,\widetilde{r}}^\pi(s) = \sum_{a \in \mathcal{A}} \pi(a|s)\widehat{Q}_{h,\widetilde{r}}^\pi(s, a)$

                $\widehat{Q}_{h,\underline{c}}^\pi(s, a) = \max\{c_h(s, a) - b_{h,c}^{k,t,\pi}(s, a) + \widehat{P}_{s,a,h}\widehat{V}_{h+1,\underline{c}}^\pi, 0\}$

                $\widehat{V}_{h,\underline{c}}^\pi(s) = \sum_{a \in \mathcal{A}} \pi(a|s)\widehat{Q}_{h,\underline{c}}^\pi(s, a)$

**return** $\bar{\pi} = \frac{1}{K} \sum_{k=1}^K \pi^k$

---

Finally, we state our algorithm in algorithm 1. For each episode, we first execute $T$ iterations of primal and dual updates. The agent executes the mixture policy $\pi^k$ and receives the reward, cost, and the next state. We update the empirical model using the doubling technique as described before. In the end, the algorithm outputs the mixture policy $\bar{\pi}$ that mixes all intermediate policies.

## 5 Main Results and Analysis

We first present the regret and constraint violation bounds of our algorithm and proofs, while we leave intermediate lemmas and proofs used to support the main results in the appendix.

## 5.1 Regret and constraint violation results

**Theorem 1** (Regret bound of algorithm 1 for relaxed feasibility). *Let $b' = b + \tau$ in algorithm 1 for some $\tau > 0$. With probability at least $1 - \delta$, the regret of algorithm 1 is*

$$Regret(K) = \widetilde{O}\left(\sqrt{SAH^3K} + 2\varepsilon_1 HK\sqrt{T} + \frac{H^2K}{\tau\sqrt{T}}\right).$$

*Proof.* Recall the definition of regret: $Regret(K) = \sum_{k=1}^{K}\left(V_{1,r}^*(s_1^k) - V_{1,r}^{\pi^k}(s_1^k)\right)$. Note that $\pi^*$ is the optimal solution to the original CMDP optimization problem eq. (1), while for each episode $\pi^k$ is an approximation solution to the empirical CMDP optimization problem eq. (6). To cope with the gap between the two policies, we introduce a proxy policy $\pi^{\tau,*}$ defined as the optimal solution to the following optimization problem

$$\pi^{\tau,*} \in \arg\max_{\pi} V_{1,r}^\pi(s_1^k), \quad \text{s.t.} \quad V_{1,c}^\pi(s_1^k) \le b' = b + \tau. \tag{10}$$

We decompose the regret as

$$Regret(K) = \sum_{k=1}^{K}\left(V_{1,r}^*(s_1^k) - V_{1,r}^{\pi^{\tau,*}}(s_1^k)\right) + \sum_{k=1}^{K}\left(V_{1,r}^{\pi^{\tau,*}}(s_1^k) - \widehat{V}_{1,\widetilde{r}}^{\pi^{\tau,*}}(s_1^k)\right)$$

$$+ \sum_{k=1}^{K}\left(\widehat{V}_{1,\widetilde{r}}^{\pi^{\tau,*}}(s_1^k) - \widehat{V}_{1,\widetilde{r}}^{\pi^k}(s_1^k)\right) + \sum_{k=1}^{K}\left(\widehat{V}_{1,\widetilde{r}}^{\pi^k}(s_1^k) - V_{1,r}^{\pi^k}(s_1^k)\right),$$

and give the regret bound by bounding each term above. The first term is the error incurred by replacing the original constraint constant $b$ by a relaxed empirical constraint constant $b' = b + \tau$ for each episode $k$. Note that by definition of $\pi^*$, we have $V_{1,c}^*(s_1) \le b \le b'$. Thus by definition of $\pi^{\tau,*}$ in eq. (10), we have $V_{1,r}^{\pi^{\tau,*}}(s_1) \ge V_{1,r}^*(s_1)$. Hence, we bound the first term by 0: $\sum_{k=1}^{K}(V_{1,r}^*(s_1^k) - V_{1,r}^{\pi^{\tau,*}}(s_1^k)) \le 0$.

By definition of the proxy policy $\pi^{\tau,*}$ in eq. (10), and noting that $\tau$ is a predetermined constant, we see that $\pi^{\tau,*}$ is a fixed policy that is independent of the online learning process. Thus we can apply lemma 16 and bound the second term by 0: $\sum_{k=1}^{K}\left(V_{1,r}^{\pi^{\tau,*}}(s_1^k) - \widehat{V}_{1,\widetilde{r}}^{\pi^{\tau,*}}(s_1^k)\right) \le 0$.

The third term is the optimization error, and it is incurred because $\pi^k$ is an approximation solution generated by iterative primal-dual updates. We bound this term by using the primal update rules in lemma 2 and also by lemmas 11 and 12, we have

$$\sum_{k=1}^{K}\left(\widehat{V}_{1,\widetilde{r}}^{\pi^{\tau,*}}(s_1^k) - \widehat{V}_{1,\widetilde{r}}^{\pi^k}(s_1^k)\right) \le 2\varepsilon_1 HK\sqrt{T} + \frac{UHK}{\sqrt{T}} \le 2\varepsilon_1 HK\sqrt{T} + \frac{H^2K}{\tau\sqrt{T}}.$$

Finally, the last term in the regret decomposition is the model prediction error, consisting of the errors caused by inaccurate empirical models and additional bonus terms. Worth mentioning, this term is essentially similar to the entire regret in [15] as the algorithms share the similar exploration bonus and update rules for transition models. We state in lemma 3 the bound with probability $1 - \delta$,

$$\sum_{k=1}^{K}\left(\widehat{V}_{1,\widetilde{r}}^{\pi^k}(s_1^k) - V_{1,r}^{\pi^k}(s_1^k)\right) = \widetilde{O}(\sqrt{SAH^3K}). \tag{11}$$

Putting everything together, we conclude our desired regret bound. $\square$

**Theorem 2** (Regret bound of algorithm 1 for strict feasibility). *Let $b' = b - \Delta$ in algorithm 1 for some $\Delta > 0$. With probability at least $1 - \delta$, the regret of algorithm 1 is*

$$Regret(K) = \widetilde{O}(\sqrt{SAH^3K} + \frac{\Delta}{\zeta}HK + 2\varepsilon_1 HK\sqrt{T} + \frac{H^2K}{(\zeta - \Delta)\sqrt{T}}).$$

*Proof.* The proof to theorem 2 follows the similar idea as the proof to theorem 1. Again, we introduce a proxy policy $\pi^{\Delta,*}$ defined as the optimal solution to the following optimization problem

$$\pi^{\Delta,*} \in \arg\max_{\pi} V_{1,r}^{\pi}(s_1^k), \quad \text{s.t.} \quad V_{1,c}^{\pi}(s_1^k) \leq b' = b - \Delta. \tag{12}$$

We decompose the regret as

$$\text{Regret}(K) = \sum_{k=1}^{K} \left( V_{1,r}^{*}(s_1^k) - V_{1,r}^{\pi^{\Delta,*}}(s_1^k) \right) + \sum_{k=1}^{K} \left( V_{1,r}^{\pi^{\Delta,*}}(s_1^k) - \widehat{V}_{1,\widetilde{r}}^{\pi^{\Delta,*}}(s_1^k) \right)$$

$$+ \sum_{k=1}^{K} \left( \widehat{V}_{1,\widetilde{r}}^{\pi^{\Delta,*}}(s_1^k) - \widehat{V}_{1,\widetilde{r}}^{\pi^k}(s_1^k) \right) + \sum_{k=1}^{K} \left( \widehat{V}_{1,\widetilde{r}}^{\pi^k}(s_1^k) - V_{1,r}^{\pi^k}(s_1^k) \right),$$

and give the regret bound by bounding each term above. The first term is the error incurred by replacing the original constraint constant $b$ by a more restrictive empirical constraint constant $b' = b - \Delta$ for each episode $k$. We bound the first term in lemma 1:

$$\sum_{k=1}^{K} \left( V_{1,r}^{*}(s_1^k) - V_{1,r}^{\pi^{\Delta,*}}(s_1^k) \right) \leq \frac{\Delta}{\zeta} HK. \tag{13}$$

For the second term, we use the similar argument as in the proof to theorem 1. By definition of the proxy policy $\pi^{\Delta,*}$ in eq. (12), and noting that $\Delta$ is a predetermined constant, we see that $\pi^{\Delta,*}$ is a fixed policy that is independent of the online learning process. Thus we can apply lemma 16 and bound the second term by 0: $\sum_{k=1}^{K} \left( V_{1,r}^{\pi^{\Delta,*}}(s_1^k) - \widehat{V}_{1,\widetilde{r}}^{\pi^{\Delta,*}}(s_1^k) \right) \leq 0$.

The last two terms can be bounded with the same argument as in the proof to theorem 1. Putting everything together, we conclude the regret bound in the strict feasibility setting. $\qquad\square$

**Theorem 3** (Constraint violation bound of algorithm 1 for relaxed feasibility). *Let $b' = b + \tau$ in algorithm 1 for some $\tau > 0$. With probability at least $1 - \delta$, the constraint violation of algorithm 1 is*

$$CV(K) = \widetilde{O}(\sqrt{SAH^3K} + \frac{2\varepsilon_1 H\sqrt{T} + UH/\sqrt{T}}{U - H/\tau} K + K\tau).$$

*Proof.* Recall the definition of constraint violation and we decompose it as follows:

$$CV(K) = \left( \sum_{k=1}^{K} V_{1,c}^{\pi^k}(s_1^k) - b \right)_{+}$$

$$= \left( \sum_{k=1}^{K} \left( V_{1,c}^{\pi^k}(s_1^k) - \widehat{V}_{1,\underline{c}}^{\pi^k}(s_1^k) \right) + \sum_{k=1}^{K} \left( \widehat{V}_{1,\underline{c}}^{\pi^k}(s_1^k) - b' \right) + \sum_{k=1}^{K} (b' - b) \right)_{+}.$$

For the first term, by definition of optimistically biased estimates of rewards and cost, we note that the analysis of bounding $\sum_{k} V_{1,c}^{\pi^k}(s_1^k) - \widehat{V}_{1,\underline{c}}^{\pi^k}(s_1^k)$ and $\sum_{k} \widehat{V}_{1,\widetilde{r}}^{\pi^k}(s_1^k) - V_{1,r}^{\pi^k}(s_1^k)$ are analogous, and mostly identical. Hence, by lemma 3, we have

$$\sum_{k=1}^{K} \left( V_{1,c}^{\pi^k}(s_1^k) - \widehat{V}_{1,\underline{c}}^{\pi^k}(s_1^k) \right) \leq \widetilde{O}(\sqrt{SAH^3K}), \tag{14}$$

with probability at least $1 - SAHK\delta'$.

The second term is the optimization error in the primal-dual process. We calculate $\pi^k$ as an approximate solution to the empirical optimization problem defined in eq. (6). Thus, it is not necessarily satisfied that $\widehat{V}_{1,\underline{c}}^{\pi^k}(s_1^k) \leq b'$. We hence return to the analysis of the primal-dual framework, and adapt techniques used in [26, 18]. By lemmas 11 to 14, we have

$$\sum_{k=1}^{K} \left( \widehat{V}_{1,\underline{c}}^{\pi^k}(s_1^k) - b' \right) \leq \sum_{k=1}^{K} \left( \widehat{V}_{1,\underline{c}}^{\pi^k}(s_1^k) - b' \right)_{+} \leq \frac{2\varepsilon_1 H\sqrt{T} + UH/\sqrt{T}}{U - H/\tau} K,$$

For the third term, we recall the definition of $b'$, and we have $\sum_{k=1}^{K} (b' - b) = K\tau$. Finally, putting everything together, we have the desired upper bound on constraint violation. $\qquad\square$

**Theorem 4** (Constraint violation bound of algorithm 1 for strict feasibility). *Let $b' = b - \Delta$ for some $\Delta > 0$. With probability at least $1 - \delta$, the constraint violation of algorithm 1 is*

$$CV(K) = \widetilde{O}(\sqrt{SAH^3 K} + \frac{2\varepsilon_1 H\sqrt{T} + UH/\sqrt{T}}{U - H/(\zeta - \Delta)} K - K\Delta).$$

*Proof.* We decompose the violation similarly as in the proof to theorem 3 and note that the first two terms can be bounded using the same argument. Then, recall $b' = b - \Delta$, and we immediately have the upper bound on constraint violation for strict feasibility. $\square$

## 5.2 Sample complexity bounds

We present the sample complexity bounds of algorithm 1. We derive these bounds by converting them from the regret and constraint violation upper bounds presented above.

**Theorem 5** (Sample complexity of algorithm 1 for relaxed feasibility). *For a fixed $\varepsilon \in (0, H]$ and $\delta \in (0, 1)$, with $K = \widetilde{O}\left(\frac{SAH^3}{\varepsilon^2}\right)$, $T = O\left(\frac{H^4}{\varepsilon^4}\right)$, $U = O\left(\frac{H}{\varepsilon}\right)$, $\varepsilon_1 = O\left(\frac{\varepsilon^3}{H^3}\right)$, and $\tau = \frac{\varepsilon}{2}$, algorithm 1 returns a policy $\bar{\pi}$ that satisfies eq. (3) with probability $1 - \delta$.*

*Proof.* Note that $\bar{\pi} = \frac{1}{K} \sum_{k=1}^{K} \pi^k$ is a mixture policy, and we have

$$V_{1,r}^*(s_1) - V_{1,r}^{\bar{\pi}}(s_1) = \frac{1}{K} \sum_{k=1}^{K} \left(V_{1,r}^*(s_1) - V_{1,r}^{\pi^k}(s_1)\right) = \frac{\text{Regret}(K)}{K},$$

and

$$V_{1,c}^{\bar{\pi}}(s_1) - b = \frac{1}{K} \sum_{k=1}^{K} \left(V_{1,c}^{\pi^k}(s_1) - b\right) \leq \frac{CV(K)}{K}.$$

By setting $K = \widetilde{O}\left(\frac{SAH^3}{\varepsilon^2}\right)$, $T = O\left(\frac{H^4}{\varepsilon^4}\right)$, $U = O\left(\frac{H}{\varepsilon}\right)$, $\varepsilon_1 = O\left(\frac{\varepsilon^3}{H^3}\right)$, and $\tau = \frac{\varepsilon}{2}$, with proper coefficients we have $V_{1,r}^*(s_1) - V_{1,r}^{\bar{\pi}}(s_1) \leq \varepsilon$, and $V_{1,c}^{\bar{\pi}}(s_1) - b \leq \varepsilon$. $\square$

Theorem 5 shows the sample complexity of algorithm 1 is $\widetilde{O}\left(\frac{SAH^3}{\varepsilon^2}\right)$ in the relaxed feasibility setting. For unconstrained MDPs with online access, [11] provided a lower bound of $\Omega\left(\frac{SAH^3}{\varepsilon^2}\right)$ for $\varepsilon$-optimal policy identification. We thus conclude that learning online CMDPs in the relaxed feasibility setting is as easy as learning unconstrained MDPs.

**Theorem 6** (Sample complexity of algorithm 1 for strict feasibility). *For a fixed $\varepsilon \in (0, H - \zeta]$ and $\delta \in (0, 1)$, with $K = \widetilde{O}\left(\frac{SAH^5}{\varepsilon^2 \zeta^2}\right)$, $T = O\left(\frac{H^6}{\zeta^4 \varepsilon^2}\right)$, $U = O\left(\frac{H^2}{\zeta(H-\varepsilon)}\right)$, $\varepsilon_1 = O\left(\frac{\varepsilon^2 \zeta^2}{H^4}\right)$, and $\Delta = \frac{\zeta\varepsilon}{2H}$, algorithm 1 returns a policy $\bar{\pi}$ that satisfies eq. (4) with probability $1 - \delta$.*

*Proof.* The proof is mostly the same as the proof to theorem 5:

$$V_{1,r}^*(s_1) - V_{1,r}^{\bar{\pi}}(s_1) = \frac{1}{K} \sum_{k=1}^{K} \left(V_{1,r}^*(s_1) - V_{1,r}^{\pi^k}(s_1)\right) = \frac{\text{Regret}(K)}{K},$$

and

$$V_{1,c}^{\bar{\pi}}(s_1) - b = \frac{1}{K} \sum_{k=1}^{K} \left(V_{1,c}^{\pi^k}(s_1) - b\right) \leq \frac{CV(K)}{K}.$$

By setting $K = \widetilde{O}\left(\frac{SAH^5}{\varepsilon^2 \zeta^2}\right)$, $T = O\left(\frac{H^6}{\zeta^4 \varepsilon^2}\right)$, $U = O\left(\frac{H^2}{\zeta(H-\varepsilon)}\right)$, $\varepsilon_1 = O\left(\frac{\varepsilon^2 \zeta^2}{H^4}\right)$, and $\Delta = \frac{\zeta\varepsilon}{2H}$, with proper coefficients, we have $V_{1,r}^*(s_1) - V_{1,r}^{\bar{\pi}}(s_1) \leq \varepsilon$, and $V_{1,c}^{\bar{\pi}}(s_1) - b \leq 0$. $\square$

Theorem 6 shows the sample complexity of algorithm 1 is $\widetilde{O}\left(\frac{SAH^5}{\varepsilon^2\zeta^2}\right)$ in the strict feasibility setting. With access to a generative model, [18] provided a lower bound of $\Omega\left(\frac{SA}{(1-\gamma)^5\varepsilon^2\zeta^2}\right)$ for infinite-horizon discounted CMDPs, which translates to $\Omega\left(\frac{SAH^5}{\varepsilon^2\zeta^2}\right)$ for episodic CMDPs. Note that this lower bound can be readily applied to learning with online access: Suppose there exists an online algorithm $\mathscr{A}$ achieving better sample complexity than the lower bound for the random access setting. Since trajectories generated in the online setting can be simulated with a generative model, we can easily construct a sampling scheme with the generative model using $\mathscr{A}$ as a subroutine. This contradicts the lower bound. We thus conclude that in the strict feasibility setting, learning CMDPs with online access is as easy as learning CMDPs with random access.

## Acknowledgement

Chang Liu, Yunfan Li, and Lin F. Yang are supported in part by NSF Grant 2221871 and an Amazon Faculty Award.

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

# A Regret Analysis

**Lemma 1.** *Let $\pi^{\Delta,*}$ be defined as in eq. (12), then*

$$\sum_{k=1}^{K} \left( V_{1,r}^*(s_1^k) - V_{1,r}^{\pi^{\Delta,*}}(s_1^k) \right) \leq \frac{\Delta}{\zeta} HK.$$

*Proof.* Recall the definition

$$\pi_c^* = \arg\max_{\pi} b - V_{1,c}^{\pi}(s_1),$$

and $\zeta = b - V_{1,c}^{\pi_c^*}(s_1)$. For each episode $k$, we define a fixed policy $\mathring{\pi}^{\Delta} = (1 - \frac{\Delta}{\zeta})\pi^* + \frac{\Delta}{\zeta}\pi_c^*$, and its value function satisfies

$$V_{1,c}^{\mathring{\pi}^{\Delta}}(s_1) = (1 - \frac{\Delta}{\zeta})V_{1,c}^*(s_1) + \frac{\Delta}{\zeta}V_{1,c}^{\pi_c^*}(s_1) \leq (1 - \frac{\Delta}{\zeta})b + \frac{\Delta}{\zeta}(b - \zeta) = b - \Delta.$$

Then,

$$\sum_{k=1}^{K} V_{1,r}^*(s_1^k) - V_{1,r}^{\pi^{\Delta,*}}(s_1^k)$$

$$\leq \sum_{k=1}^{K} V_{1,r}^*(s_1^k) - V_{1,r}^{\mathring{\pi}^{\Delta}}(s_1^k)$$

$$= \sum_{k=1}^{K} V_{1,r}^*(s_1^k) - \left( (1 - \frac{\Delta}{\zeta})V_{1,r}^*(s_1^k) + \frac{\Delta}{\zeta}V_{1,r}^{\pi_c^*}(s_1^k) \right)$$

$$= \sum_{k=1}^{K} \frac{\Delta}{\zeta}(V_{1,r}^*(s_1^k) - V_{1,r}^{\pi_c^*}(s_1^k))$$

$$\leq \frac{\Delta}{\zeta} HK,$$

where the first inequality is due to the definition of $\pi^{\Delta,*}$, i.e., for any policy $\pi$, s.t. $V_{1,c}^{\pi}(s_1) \leq b' = b - \Delta$, $V_{1,r}^{\pi^{\Delta,*}}(s_1) \geq V_{1,r}^{\pi}(s_1)$, and the second inequality follows from $V_{1,r}^*(s_1) \leq H$. $\qquad\square$

**Lemma 2.** *Let $\pi^k$ be the mixture policy in algorithm 1, $\pi^{\tau,*}$ and $\pi^{\Delta,*}$ be defined as in eqs. (10) and (12), then*

$$\sum_{k=1}^{K} \left( \widehat{V}_{1,\widetilde{r}}^{\pi^{\tau,*}}(s_1^k) - \widehat{V}_{1,\widetilde{r}}^{\pi^k}(s_1^k) \right) \leq 2\varepsilon_1 H\sqrt{T} + \frac{UH}{\sqrt{T}},$$

*and*

$$\sum_{k=1}^{K} \left( \widehat{V}_{1,\widetilde{r}}^{\pi^{\Delta,*}}(s_1^k) - \widehat{V}_{1,\widetilde{r}}^{\pi^k}(s_1^k) \right) \leq 2\varepsilon_1 H\sqrt{T} + \frac{UH}{\sqrt{T}}.$$

*Proof.* For any primal-dual iteration $t \in [T]$,

$$\widehat{V}_{1,\widetilde{r}}^{\pi^{\Delta,*}}(s_1^k) - \widehat{\lambda}_t^k \widehat{V}_{1,\underline{c}}^{\pi^{\Delta,*}}(s_1^k) \leq \widehat{V}_{1,\widetilde{r}}^{\widehat{\pi}_t^k}(s_1^k) - \widehat{\lambda}_t^k \widehat{V}_{1,\underline{c}}^{\widehat{\pi}_t^k}(s_1^k).$$

Taking average over $T$ iterations,

$$\frac{1}{T}\sum_{t=1}^{T} \left( \widehat{V}_{1,\widetilde{r}}^{\pi^{\Delta,*}}(s_1^k) - \widehat{\lambda}_t^k \widehat{V}_{1,\underline{c}}^{\pi^{\Delta,*}}(s_1^k) \right) \leq \frac{1}{T}\sum_{t=1}^{T} \left( \widehat{V}_{1,\widetilde{r}}^{\widehat{\pi}_t^k}(s_1^k) - \widehat{\lambda}_t^k \widehat{V}_{1,\underline{c}}^{\widehat{\pi}_t^k}(s_1^k) \right).$$

Note that the mixture policy $\pi^k$ is the average policies of $\widehat{\pi}_t^k$, we have

$$\widehat{V}_{1,\widetilde{r}}^{\pi^{\Delta,*}}(s_1^k) - \frac{1}{T}\sum_{t=1}^{T} \widehat{\lambda}_t^k \widehat{V}_{1,\underline{c}}^{\pi^{\Delta,*}}(s_1^k) \leq \widehat{V}_{1,\widetilde{r}}^{\pi^k}(s_1^k) - \frac{1}{T}\sum_{t=1}^{T} \widehat{\lambda}_t^k \widehat{V}_{1,\underline{c}}^{\widehat{\pi}_t^k}(s_1^k). \tag{15}$$

Further, we notice that

$$\widehat{V}_{1,\underline{c}}^{\pi^{\Delta,*}} \le V_{1,c}^{\pi^{\Delta,*}} \le b'. \tag{16}$$

Thus, for any episode $k$,

$$\widehat{V}_{1,\widetilde{r}}^{\pi^{\Delta,*}}(s_1^k) - \widehat{V}_{1,\widetilde{r}}^{\pi^k}(s_1^k)$$

$$= \left(\widehat{V}_{1,\widetilde{r}}^{\pi^{\Delta,*}}(s_1^k) - \frac{1}{T}\sum_{t=1}^{T}\widehat{\lambda}_t^k \widehat{V}_{1,\underline{c}}^{\pi^{\Delta,*}}(s_1^k)\right) - \left(\widehat{V}_{1,\widetilde{r}}^{\pi^k}(s_1^k) - \frac{1}{T}\sum_{t=1}^{T}\widehat{\lambda}_t^k \widehat{V}_{1,\underline{c}}^{\widehat{\pi}_t^k}(s_1^k)\right)$$

$$+ \frac{1}{T}\sum_{t=1}^{T}\widehat{\lambda}_t^k\left(\widehat{V}_{1,\underline{c}}^{\pi^{\Delta,*}}(s_1^k) - \widehat{V}_{1,\underline{c}}^{\widehat{\pi}_t^k}(s_1^k)\right)$$

$$\le \frac{1}{T}\sum_{t=1}^{T}\widehat{\lambda}_t^k(\widehat{V}_{1,\underline{c}}^{\pi^{\Delta,*}}(s_1^k) - \widehat{V}_{1,\underline{c}}^{\widehat{\pi}_t^k}(s_1^k))$$

$$\le \frac{1}{T}\sum_{t=1}^{T}\widehat{\lambda}_t^k(b' - \widehat{V}_{1,\underline{c}}^{\widehat{\pi}_t^k}(s_1^k))$$

$$\le 2\varepsilon_1 H\sqrt{T} + \frac{UH}{\sqrt{T}},$$

where the first inequality follows from eq. (15), the second inequality follows from eq. (16), and the last inequality follows from lemma 14 by letting $\lambda = 0$ The proof for $\pi^{\tau,*}$ is analogous and is thus omitted. $\square$

**Lemma 3.**

$$\sum_{k=1}^{K}\left(\widehat{V}_{1,\widetilde{r}}^{\pi^k}(s_1^k) - V_{1,r}^{\pi^k}(s_1^k)\right) = \widetilde{O}(\sqrt{SAH^3K}).$$

*Proof.* By definition, we write

$$\widehat{V}_{h,\widetilde{r}}^{\pi^k}(s_h^k) = \sum_{a\in\mathcal{A}}\pi^k(a|s_h^k)\widehat{Q}_{h,\widetilde{r}}^{\pi^k}(s_h^k, a)$$

$$= \widehat{Q}_{h,\widetilde{r}}^{\pi^k}(s_h^k, a_h^k) + \left(\sum_{a\in\mathcal{A}}\pi^k(a|s_h^k)\widehat{Q}_{h,\widetilde{r}}^{\pi^k}(s_h^k, a) - \widehat{Q}_{h,\widetilde{r}}^{\pi^k}(s_h^k, a_h^k)\right)$$

$$\le r_h(s_h^k, a_h^k) + b_h^k(s_h^k, a_h^k) + \widehat{P}_{s_h^k,a_h^k,h}^k \widehat{V}_{h+1,\widetilde{r}}^{\pi^k} + \phi_h^k$$

$$\le r_h(s_h^k, a_h^k) + b_h^k(s_h^k, a_h^k) + (\widehat{P}_{s_h^k,a_h^k,h}^k - P_{s,a,h})\widehat{V}_{h+1,\widetilde{r}}^{\pi^k} + (P_{s_h^k,a_h^k,h} - \mathbb{1}_{\{s_{h+1}^k\}})\widehat{V}_{h+1,\widetilde{r}}^{\pi^k}$$

$$+ \widehat{V}_{h+1,\widetilde{r}}^{\pi^k}(s_{h+1}^k) + \phi_h^k,$$

where

$$\phi_h^k = \left(\sum_{a\in\mathcal{A}}\pi^k(a|s_h^k)\widehat{Q}_{h,\widetilde{r}}^{\pi^k}(s_h^k, a) - \widehat{Q}_{h,\widetilde{r}}^{\pi^k}(s_h^k, a_h^k)\right)$$

is a zero-mean random variable conditional on $\pi^k$. Then by summing over $H$ time steps and telescoping, we have

$$\widehat{V}_{1,\widetilde{r}}^{\pi^k}(s_1^k) \le \sum_{h=1}^{H} r_h(s_h^k, a_h^k) + b_h^k(s_h^k, a_h^k) + (\widehat{P}_{s_h^k,a_h^k,h}^k - P_{s,a,h})\widehat{V}_{h+1,\widetilde{r}}^{\pi^k}$$

$$+ (P_{s_h^k,a_h^k,h} - \mathbb{1}_{\{s_{h+1}^k\}})\widehat{V}_{h+1,\widetilde{r}}^{\pi^k} + \sum_{h=1}^{H}\phi_h^k.$$

The term we want to bound is now decomposed as

$$\sum_{k=1}^{K}\left(\widehat{V}_{1,\widetilde{r}}^{\pi^k}(s_1^k) - V_{1,r}^{\pi^k}(s_1^k)\right) \le \sum_{k=1}^{K}\sum_{h=1}^{H} b_h^k(s_h^k, a_h^k) + \sum_{k=1}^{K}\sum_{h=1}^{H}(\widehat{P}_{s_h^k, a_h^k, h}^k - P_{s,a,h})\widehat{V}_{h+1,\widetilde{r}}^{\pi^k} + \sum_{k=1}^{K}\sum_{h=1}^{H}\phi_h^k$$

$$+ \sum_{k=1}^{K}\sum_{h=1}^{H}(P_{s_h^k, a_h^k, h} - \mathbb{1}_{\{s_{h+1}^k\}})\widehat{V}_{h+1,\widetilde{r}}^{\pi^k} + \sum_{k=1}^{K}\left(\sum_{h=1}^{H} r_h(s_h^k, a_h^k) - V_{1,r}^{\pi^k}(s_1^k)\right).$$

We apply lemmas 4, 5 and 7 to 9, and conlude that with probability $1 - \delta$,

$$\sum_{k=1}^{K}\left(\widehat{V}_{1,\widetilde{r}}^{\pi^k}(s_1^k) - V_{1,r}^{\pi^k}(s_1^k)\right) = O\left(\sqrt{SAH^3 K \log^5 \frac{SAHK}{\delta}}\right).$$

$\square$

**Lemma 4.** *With probability at least* $1 - 3SAHK\delta'$,

$$\sum_{k=1}^{K}\sum_{h=1}^{H} b_{h,r}^{k,\pi^k}(s_h^k, a_h^k) \le \widetilde{O}(\sqrt{SAH^3 K}).$$

*Proof.* By definition of bonus $b_{h,r}^{k,\pi^k}(s_h^k, a_h^k)$, we have

$$\sum_{k=1}^{K}\sum_{h=1}^{H}, b_{h,r}^{k,\pi^k}(s_h^k, a_h^k) = \frac{460}{9}\sum_{k,h}\sqrt{\frac{\mathbb{V}\left(\widehat{P}_{s_h^k, a_h^k, h}^k, \widehat{V}_{h+1,\widetilde{r}}^{\pi^k}\right)\log\frac{1}{\delta'}}{N_h^k(s_h^k, a_h^k)}} + \frac{544}{9}\sum_{k,h}\frac{H\log\frac{1}{\delta'}}{N_h^k(s_h^k, a_h^k)}.$$

Applying the Cauchy-Schwarz inequality and lemma 17, we obtain

$$\sum_{k=1}^{K}\sum_{h=1}^{H} b_{h,r}^{k,\pi^k}(s_h^k, a_h^k) \le \frac{460}{9}\sqrt{\sum_{k,h}\frac{\log\frac{1}{\delta'}}{N_h^k(s_h^k, a_h^k)}}\sqrt{\sum_{k,h}\mathbb{V}\left(\widehat{P}_{s_h^k, a_h^k, h}^k, \widehat{V}_{h+1,\widetilde{r}}^{\pi^k}\right)}$$

$$+ \frac{544H\log\frac{1}{\delta'}}{9}\sum_{k,h}\frac{1}{N_h^k(s_h^k, a_h^k)}$$

$$\le \frac{460}{9}\sqrt{2SAH(\log_2 K)\left(\log\frac{1}{\delta'}\right)\sum_{k,h}\mathbb{V}\left(\widehat{P}_{s_h^k, a_h^k, h}^k, \widehat{V}_{h+1,\widetilde{r}}^{\pi^k}\right)}$$

$$+ \frac{1088}{9}SAH^2(\log_2 K)\log\frac{1}{\delta'}.$$

Then by lemma 10, we have the desired result. $\square$

**Lemma 5.**

$$\sum_{k=1}^{K}\sum_{h=1}^{H}(\widehat{P}_{s_h^k, a_h^k, h}^k - P_{s,a,h})\widehat{V}_{h+1,\widetilde{r}}^{\pi^k} \le \widetilde{O}(\sqrt{SAH^3 K}).$$

*Proof.* First we notice that $\pi^k$ is a mixture policy of $\pi^k = \frac{1}{T}\sum_{t=1}^{T}\widehat{\pi}_t^k$, then we have

$$\sum_{k=1}^{K}\sum_{h=1}^{H}(\widehat{P}_{s_h^k, a_h^k, h}^k - P_{s,a,h})\widehat{V}_{h+1,\widetilde{r}}^{\pi^k} = \sum_{k=1}^{K}\sum_{h=1}^{H}(\widehat{P}_{s_h^k, a_h^k, h}^k - P_{s,a,h})\frac{1}{T}\sum_{t=1}^{T}\widehat{V}_{h+1,\widetilde{r}}^{\widehat{\pi}_t^k}$$

$$= \frac{1}{T}\sum_{t=1}^{T}\sum_{k=1}^{K}\sum_{h=1}^{H}(\widehat{P}_{s_h^k, a_h^k, h}^k - P_{s,a,h})\widehat{V}_{h+1,\widetilde{r}}^{\widehat{\pi}_t^k}.$$

Note that given a total profile $\mathcal{I} \in \mathcal{C}$ and a dual variable sequence $\lambda \in \Lambda$, $\widehat{V}_{h+1,\widetilde{r}}^{\widehat{\pi}_t^k}$ is determined by

$$\left\{\widehat{P}_{s,a,h'}^{(I_{s,a,h'}^k)}, r_{h'}^{(I_{s,a,h'}^k)}(s,a), c_{h'}^{(I_{s,a,h'}^k)}(s,a)\right\}_{h < h' \le H, (s,a,k) \in \mathcal{S} \times \mathcal{A} \times [K]},$$

and $\|\widehat{V}_{h+1,\widetilde{r}}^{\pi^k}\|_\infty \le H$. Thus we can invoke lemma 6 and also by lemma 10, we have

$$\sum_{k=1}^K \sum_{h=1}^H (\widehat{P}_{s_h^k,a_h^k,h}^k - P_{s,a,h})\widehat{V}_{h+1,\widetilde{r}}^{\pi^k} \le \widetilde{O}(\sqrt{SAH^3K}).$$

$\square$

**Lemma 6.** *Let us first specify the types of vectors $\{X_{h,s,a}\}$. For each total profile $\mathcal{I} \in \mathcal{C}$, consider any set $\{\mathcal{X}_{h,\mathcal{I}}\}_{1 \le h \le H}$ obeying: for each $1 \le h \le H$,*

- $\mathcal{X}_{h+1,\mathcal{I}}$ *is given by a deterministic function of $\mathcal{I}$, a dual variable $\lambda \in \Lambda$, and*

$$\left\{ \widehat{P}_{s,a,h'}^{(I_{s,a,h'}^k)}, r_{h'}^{(I_{s,a,h'}^k)}(s,a), c_{h'}^{(I_{s,a,h'}^k)}(s,a) \right\}_{h<h'\le H, (s,a,k)\in \mathcal{S}\times\mathcal{A}\times[K]};$$

- $\|X\|_\infty \le H$ *for each vector $X \in \mathcal{X}_{h,\mathcal{I}}$;*

- $\mathcal{X}_{h,\mathcal{I}}$ *is a set of no more than $K+1$ non-negative vectors in $\mathbb{R}^S$, and contains the all-zero vector 0.*

*Suppose that $K \ge SAH\log_2 K$, and construct a set $\{\mathcal{X}_{h,\mathcal{I}}\}_{1 \le h \le H}$ for each $\mathcal{I} \in \mathcal{C}$ satisfying the above properties. Then with probability at least $1 - \delta'$,*

$$\sum_{s,a,h\in\mathcal{S}\times\mathcal{A}\times[H]} \left\langle \widehat{P}_{s,a,h}^{(l)} - P_{s,a,h}, X_{h+1,s,a} \right\rangle \le \sum_{s,a,h\in\mathcal{S}\times\mathcal{A}\times[H]} \max\left\{ \left\langle \widehat{P}_{s,a,h}^{(l)} - P_{s,a,h}, X_{h+1,s,a} \right\rangle, 0 \right\}$$

$$\le \sqrt{\frac{8}{2^{l-2}} \sum_{s,a,h} \mathbb{V}(P_{s,a,h}, X_{h+1,s,a})\left(6SAH\log_2^2 K\right)} + \frac{4H}{2^{l-2}}\left(6SAH\log_2^2 K\right)$$

*holds simultaneously for all $\mathcal{I} \in \mathcal{C}$, all dual variable sequences, all $2 \le l \le \log_2 K + 1$, and all sequences $\{X_{h,s,a}\}_{(s,a,h)\in\mathcal{S}\times\mathcal{A}\times[H]}$ obeying $X_{h,s,a} \in \mathcal{X}_{h+1,\mathcal{I}}, \forall (s,a,h) \in \mathcal{S} \times \mathcal{A} \times [H]$.*

*Proof.* This proof is adapted from the proof to lemma 6 in [15]. Let us begin by considering any fixed total profile $\mathcal{I} \in \mathcal{C}$, any fixed integer $l$ obeying $2 \le l \le \log_2 K + 1$, and any given feasible sequence $\{X_{h,s,a}\}_{(s,a,h)\in\mathcal{S}\times\mathcal{A}\times[H]}$. Recall that (i) $\widehat{P}_{s,a,h}^{(l)}$ is computed based on the $l$-th batch of data comprising $2^{l-2}$ independent samples; and (ii) each $X_{h+1,s,a}$ is given by a deterministic function of $\mathcal{I}$ and the empirical models for steps $h' \in [h+1, H]$. Consequently, lemma 18 tells us that: with probability at least $1 - \delta'$, one has

$$\sum_{s,a,h} \left\langle \widehat{P}_{s,a,h}^{(l)} - P_{s,a,h}, X_{h+1,s,a} \right\rangle$$

$$\le \sqrt{\frac{8}{2^{l-2}} \sum_{s,a,h} \mathbb{V}(P_{s,a,h}, X_{h+1,s,a}) \log\frac{3\log_2(SAHK)}{\delta'}} + \frac{4H}{2^{l-2}} \log\frac{3\log_2(SAHK)}{\delta'}$$

where we view the left-hand side as a martingale sequence from $h = H$ back to $h = 1$. Moreover, given that each $X_{h,s,a}$ has at most $K + 1$ different choices (since we assume $|\mathcal{X}_{h,\mathcal{I}}| \le K+1$), there are no more than $(K + 1)^{SAH} \le (2K)^{SAH}$ possible choices of the feasible sequence $\{X_{h,s,a}\}_{(s,a,h)\in\mathcal{S}\times\mathcal{A}\times[H]}$. In addition, it has been shown in Lemma 5 of [15] that there are no more than $(4SAHK)^{2SAH}\log_2 K$ possibilities of the total profile $\mathcal{I}$. Taking the union bound over all these choices and replacing $\delta'$ with $\delta'/\left((4SAHK)^{2SAH}\log_2 K(2K)^{SAH}\log_2 K|\Lambda|KT\right)$, we can demonstrate that with probability at least $1 - \delta'$,

$$\sum_{s,a,h} \left\langle \widehat{P}_{s,a,h}^{(l)} - P_{s,a,h}, X_{h+1,s,a} \right\rangle$$

$$\le \sqrt{\frac{8}{2^{l-2}} \sum_{s,a,h} \mathbb{V}(P_{s,a,h}, X_{h+1,s,a})\left(6SAH\log_2^2 K\right)} + \frac{4H}{2^{l-2}}\left(6SAH\log_2^2 K\right)$$

holds simultaneously for all $\mathcal{I} \in \mathcal{C}$, all dual variable sequences, all $2 \leq l \leq \log_2 K + 1$, and all feasible sequences $\{X_{h,s,a}\}_{(s,a,h) \in \mathcal{S} \times \mathcal{A} \times [H]}$. Finally, recalling our assumption $0 \in \mathcal{X}_{h+1,\mathcal{I}}$, we see that for every total profile $\mathcal{I}$ and its associated feasible sequence $\{X_{h,s,a}\}$

$$\sum_{s,a,h} \max \left\{ \left\langle \widehat{P}_{s,a,h}^{(l)} - P_{s,a,h}, X_{h+1,s,a} \right\rangle, 0 \right\} \in \left\{ \sum_{s,a,h} \left\langle \widehat{P}_{s,a,h}^{(l)} - P_{s,a,h}, \widetilde{X}_{h+1,s,a} \right\rangle \mid \widetilde{X}_{h+1,s,a} \in \mathcal{X}_{h+1,\mathcal{I}}, \forall (s,a,h) \right\}$$

holds true. Consequently, the uniform upper bound on the right-hand side continues to be a valid upper bound on $\sum_{s,a,h} \max \left\{ \left\langle \widehat{P}_{s,a,h}^{(l)} - P_{s,a,h}, X_{h+1,s,a} \right\rangle, 0 \right\}$. This concludes the proof. $\quad\square$

**Lemma 7.** *With probability at least* $1 - 4\delta' \log(KH)$,

$$\sum_{k=1}^{K} \sum_{h=1}^{H} \phi_h^k \leq \widetilde{O}(\sqrt{H^3 K}).$$

*Proof.* Note that $\phi_h^k = \left( \sum_{a \in \mathcal{A}} \pi^k(a|s_h^k) \widehat{Q}_{h,\widetilde{r}}^{\pi^k}(s_h^k, a) - \widehat{Q}_{h,\widetilde{r}}^{\pi^k}(s_h^k, a_h^k) \right)$ is a zero-mean random variable conditional on $\pi^k$ and is upper bounded by constant $H$. By lemma 18, we have

$$\sum_{k=1}^{K} \sum_{h=1}^{H} \phi_h^k \leq 2\sqrt{2} \sqrt{\sum_{k=1}^{K} \sum_{h=1}^{H} \mathsf{Var}(\phi_h^k) \log \frac{1}{\delta'}} + 3H \log \frac{1}{\delta'}$$

$$\leq 2 \sqrt{2KH^3 \log \frac{1}{\delta'}} + 3H \log \frac{1}{\delta'}$$

with probability at least $1 - 4\delta' \log(KH)$. $\quad\square$

**Lemma 8.** *With probability at least* $1 - SAH^2 K^2 \delta'$,

$$\sum_{k=1}^{K} \sum_{h=1}^{H} (P_{s_h^k, a_h^k, h} - \mathbf{1}_{s_{h+1}^k}) \widehat{V}_{h+1,\widetilde{r}}^{\pi^k} \leq \widetilde{O}(\sqrt{H^2 K}).$$

*Proof.* We note that conditional on state-action pair $(s_h^k, a_h^k)$, the vectors $P_{s_h^k, a_h^k, h}$ and $\mathbf{1}_{s_{h+1}^k}$ are both independent of the value function estimate $\widehat{V}_{h+1,\widetilde{r}}^{\pi^k}$. Also, the vector $\mathbf{1}_{s_{h+1}^k}$ has the mean of $P_{s_h^k, a_h^k, h}$. Hence, $(P_{s_h^k, a_h^k, h} - \mathbf{1}_{s_{h+1}^k}) \widehat{V}_{h+1,\widetilde{r}}^{\pi^k}$ is a zero-mean random variable bounded by $H$ from above, and we thus apply lemma 18 and have

$$\sum_{k=1}^{K} \sum_{h=1}^{H} (P_{s_h^k, a_h^k, h} - \mathbf{1}_{s_{h+1}^k}) \widehat{V}_{h+1,\widetilde{r}}^{\pi^k} \leq 2\sqrt{2} \sqrt{\sum_{k=1}^{K} \sum_{h=1}^{H} \mathbb{V} \left( P_{s_h^k, a_h^k, h}, \widehat{V}_{h+1,\widetilde{r}}^{\pi^k} \right) \log \frac{1}{\delta'}} + 3H \log \frac{1}{\delta'}$$

with probability at least $1 - SAH^2 K^2 \delta'$. By lemma 10, we obtain our lemma. $\quad\square$

**Lemma 9.** *With probability at least* $1 - 4\delta' \log(KH)$,

$$\sum_{k=1}^{K} \left( \sum_{h=1}^{H} r_h(s_h^k, a_h^k) - V_{1,r}^{\pi^k}(s_1^k) \right) \leq \widetilde{O}(\sqrt{H^2 K}).$$

*Proof.* Note that conditional on $\pi^k$, $E_k := \sum_{h=1}^{H} r_h(s_h^k, a_h^k) - V_{1,r}^{\pi^k}(s_1^k)$ is a zero-mean random variable upper bounded by constant $H$. By lemma 18, we have

$$\left| \sum_{k=1}^{K} E_k \right| \leq 2\sqrt{2} \sqrt{\sum_{k=1}^{K} \mathsf{Var}(E_k) \log \frac{1}{\delta'}} + 3H \log \frac{1}{\delta'}$$

$$\leq 2 \sqrt{2KH^2 \log \frac{1}{\delta'}} + 3H \log \frac{1}{\delta'},$$

with probability at least $1 - 4\delta' \log(KH)$, where the last inequality holds because $|E_k| \leq H$. $\quad\square$

**Lemma 10.** *With probability at least* $1 - 6SAHK\delta'$,

$$\sum_{k=1}^{K}\sum_{h=1}^{H} \mathbb{V}\left(\widehat{P}_{s_h^k,a_h^k,h}^{k}, \widehat{V}_{h+1,\widetilde{r}}^{\pi^k}\right) \leq \widetilde{O}(H^2 K + \sqrt{H^5 K} + SAH^3),$$

$$\sum_{k=1}^{K}\sum_{h=1}^{H} \mathbb{V}\left(P_{s_h^k,a_h^k,h}, \widehat{V}_{h+1,\widetilde{r}}^{\pi^k}\right) \leq \widetilde{O}(H^2 K + \sqrt{H^5 K} + SAH^3).$$

*Proof.* This proof is modified from the proof to lemma 11 in [15], and we show here the parts where the proofs differ. First we write by direct calculation

$$\sum_{k=1}^{K}\sum_{h=1}^{H} \mathbb{V}\left(\widehat{P}_{s_h^k,a_h^k,h}^{k}, \widehat{V}_{h+1,\widetilde{r}}^{\pi^k}\right) = \sum_{k=1}^{K}\sum_{h=1}^{H}\left(\left\langle \widehat{P}_{s_h^k,a_h^k,h}^{k}, (\widehat{V}_{h+1,\widetilde{r}}^{\pi^k})^2\right\rangle - \left\langle \widehat{P}_{s_h^k,a_h^k,h}^{k}, \widehat{V}_{h+1,\widetilde{r}}^{\pi^k}\right\rangle^2\right)$$

$$= \sum_{k=1}^{K}\sum_{h=1}^{H}\left\langle \widehat{P}_{s_h^k,a_h^k,h}^{k} - P_{s_h^k,a_h^k,h}^{k}, (\widehat{V}_{h+1,\widetilde{r}}^{\pi^k})^2\right\rangle + \sum_{k=1}^{K}\sum_{h=1}^{H}\left\langle P_{s_h^k,a_h^k,h}^{k} - \mathbf{1}_{s_{h+1}^k}, (\widehat{V}_{h+1,\widetilde{r}}^{\pi^k})^2\right\rangle$$

$$+ \sum_{k=1}^{K}\sum_{h=2}^{H}(\widehat{V}_{h,\widetilde{r}}^{\pi^k}(s_h^k))^2 - \sum_{k=1}^{K}\sum_{h=1}^{H}\left\langle \widehat{P}_{s_h^k,a_h^k,h}^{k}, \widehat{V}_{h+1,\widetilde{r}}^{\pi^k}\right\rangle^2$$

$$\leq \sum_{k=1}^{K}\sum_{h=1}^{H}\left\langle \widehat{P}_{s_h^k,a_h^k,h}^{k} - P_{s_h^k,a_h^k,h}^{k}, (\widehat{V}_{h+1,\widetilde{r}}^{\pi^k})^2\right\rangle + \sum_{k=1}^{K}\sum_{h=1}^{H}\left\langle P_{s_h^k,a_h^k,h}^{k} - \mathbf{1}_{s_{h+1}^k}, (\widehat{V}_{h+1,\widetilde{r}}^{\pi^k})^2\right\rangle$$

$$+ \sum_{k=1}^{K}\sum_{h=1}^{H}\left(\widehat{V}_{h,\widetilde{r}}^{\pi^k}(s_h^k) + \left\langle \widehat{P}_{s_h^k,a_h^k,h}^{k}, \widehat{V}_{h+1,\widetilde{r}}^{\pi^k}\right\rangle\right)\left(\widehat{V}_{h,\widetilde{r}}^{\pi^k}(s_h^k) - \left\langle \widehat{P}_{s_h^k,a_h^k,h}^{k}, \widehat{V}_{h+1,\widetilde{r}}^{\pi^k}\right\rangle\right),$$

and since the value function estimates are bounded by $H$,

$$\leq \sum_{k=1}^{K}\sum_{h=1}^{H}\left\langle \widehat{P}_{s_h^k,a_h^k,h}^{k} - P_{s_h^k,a_h^k,h}^{k}, (\widehat{V}_{h+1,\widetilde{r}}^{\pi^k})^2\right\rangle + \sum_{k=1}^{K}\sum_{h=1}^{H}\left\langle P_{s_h^k,a_h^k,h}^{k} - \mathbf{1}_{s_{h+1}^k}, (\widehat{V}_{h+1,\widetilde{r}}^{\pi^k})^2\right\rangle$$

$$+ 2H\sum_{k=1}^{K}\sum_{h=1}^{H}\max\left\{\widehat{V}_{h,\widetilde{r}}^{\pi^k}(s_h^k) - \langle \widehat{P}_{s_h^k,a_h^k,h}, \widehat{V}_{h+1,\widetilde{r}}^{\pi^k}\rangle, 0\right\}$$

$$\leq \sum_{k=1}^{K}\sum_{h=1}^{H}\left\langle \widehat{P}_{s_h^k,a_h^k,h}^{k} - P_{s_h^k,a_h^k,h}^{k}, (\widehat{V}_{h+1,\widetilde{r}}^{\pi^k})^2\right\rangle + \sum_{k=1}^{K}\sum_{h=1}^{H}\left\langle P_{s_h^k,a_h^k,h}^{k} - \mathbf{1}_{s_{h+1}^k}, (\widehat{V}_{h+1,\widetilde{r}}^{\pi^k})^2\right\rangle$$

$$+ 2H\sum_{k=1}^{K}\sum_{h=1}^{H}\max\left\{\widehat{V}_{h,\widetilde{r}}^{\pi^k}(s_h^k) - \widehat{Q}_{h,\widetilde{r}}^{\pi^k}(s_h^k, a_h^k) + \widehat{Q}_{h,\widetilde{r}}^{\pi^k}(s_h^k, a_h^k) - \langle \widehat{P}_{s_h^k,a_h^k,h}, \widehat{V}_{h+1,\widetilde{r}}^{\pi^k}\rangle, 0\right\}.$$

By definition of update rule of $\widehat{Q}$ functions, we have

$$\leq \sum_{k=1}^{K}\sum_{h=1}^{H}\left\langle \widehat{P}_{s_h^k,a_h^k,h}^{k} - P_{s_h^k,a_h^k,h}^{k}, (\widehat{V}_{h+1,\widetilde{r}}^{\pi^k})^2\right\rangle + \sum_{k=1}^{K}\sum_{h=1}^{H}\left\langle P_{s_h^k,a_h^k,h}^{k} - \mathbf{1}_{s_{h+1}^k}, (\widehat{V}_{h+1,\widetilde{r}}^{\pi^k})^2\right\rangle$$

$$+ 2H\sum_{k=1}^{K}\sum_{h=1}^{H} r_h(s_h^k, a_h^k) + 2H\sum_{k=1}^{K}\sum_{h=1}^{H} b_{h,r}^{k,\pi^k}(s_h^k, a_h^k) + 2H\sum_{k=1}^{K}\sum_{h=1}^{H}\max\{\xi_h^k, 0\},$$

where $\xi_h^k := \widehat{V}_{h,\widetilde{r}}^{\pi^k}(s_h^k) - \widehat{Q}_{h,\widetilde{r}}^{\pi^k}(s_h^k, a_h^k) = \sum_{a\in\mathcal{A}} \pi^k(a|s_h^k)\widehat{Q}_{h,\widetilde{r}}^{\pi^k}(s_h^k, a) - \widehat{Q}_{h,\widetilde{r}}^{\pi^k}(s_h^k, a_h^k)$ is a zero-mean random variable conditional on $\pi^k$ bounded by $H$. By the results of lemma 10 and 11 in [15], we finally bound

$$\sum_{k=1}^{K}\sum_{h=1}^{H} \mathbb{V}\left(\widehat{P}_{s_h^k,a_h^k,h}^{k}, \widehat{V}_{h+1,\widetilde{r}}^{\pi^k}\right) \leq \widetilde{O}(H^2 K + \sqrt{H^5 K} + SAH^3).$$

Similarly we can show that with probability at least $1 - 3SAHK\delta'$,

$$\sum_{k=1}^{K}\sum_{h=1}^{H}\mathbb{V}\left(P_{s_h^k,a_h^k,h},\widehat{V}_{h+1,\widetilde{r}}^{\pi^k}\right) = \sum_{k=1}^{K}\sum_{h=1}^{H}\left\langle P_{s_h^k,a_h^k,h},\left(V_{h+1}^k\right)^2\right\rangle - \sum_{k=1}^{K}\sum_{h=1}^{H}\left(\left\langle P_{s_h^k,a_h^k,h},V_{h+1}^k\right\rangle\right)^2$$

$$= \sum_{k=1}^{K}\sum_{h=1}^{H}\left\langle P_{s_h^k,a_h^k,h} - \mathbf{1}_{s_{h+1}^k},\left(V_{h+1}^k\right)^2\right\rangle + \sum_{k=1}^{K}\sum_{h=2}^{H}\left(V_h^k\left(s_h^k\right)\right)^2 - \sum_{k=1}^{K}\sum_{h=1}^{H}\left(\left\langle P_{s_h^k,a_h^k,h},V_{h+1}^k\right\rangle\right)^2,$$

and we invoke the similar argument as above,

$$\leq \sum_{k=1}^{K}\sum_{h=1}^{H}\left\langle P_{s_h^k,a_h^k,h} - \mathbf{1}_{s_{h+1}^k},\left(V_{h+1}^k\right)^2\right\rangle + 2H\sum_{k=1}^{K}\sum_{h=1}^{H}\max\left\{V_h^k\left(s_h^k\right) - \left\langle P_{s_h^k,a_h^k,h},V_{h+1}^k\right\rangle,0\right\}$$

$$\leq \sum_{k=1}^{K}\sum_{h=1}^{H}\left\langle P_{s_h^k,a_h^k,h} - \mathbf{1}_{s_{h+1}^k},\left(V_{h+1}^k\right)^2\right\rangle + 2H\sum_{k=1}^{K}\sum_{h=1}^{H}\max\left\{V_h^k\left(s_h^k\right) - \left\langle \widehat{P}_{s_h^k,a_h^k,h},V_{h+1}^k\right\rangle,0\right\}$$

$$+ 2H\sum_{k=1}^{K}\sum_{h=1}^{H}\max\left\{\left\langle \widehat{P}_{s_h^k,a_h^k,h}^k - P_{s_h^k,a_h^k,h},V_{h+1}^k\right\rangle,0\right\}$$

$$\leq \sum_{k=1}^{K}\sum_{h=1}^{H}\left\langle P_{s_h^k,a_h^k,h} - \mathbf{1}_{s_{h+1}^k},\left(V_{h+1}^k\right)^2\right\rangle + 2H\sum_{k=1}^{K}\sum_{h=1}^{H}r_h(s_h^k,a_h^k) + 2H\sum_{k=1}^{K}\sum_{h=1}^{H}b_{h,r}^{k,\pi^k}(s_h^k,a_h^k)$$

$$+ 2H\sum_{k=1}^{K}\sum_{h=1}^{H}\max\{\xi_h^k,0\} + 2H\sum_{k=1}^{K}\sum_{h=1}^{H}\max\left\{\left\langle \widehat{P}_{s_h^k,a_h^k,h}^k - P_{s_h^k,a_h^k,h},V_{h+1}^k\right\rangle,0\right\}$$

By the results of lemma 10 and 11 in [15], we finally bound

$$\sum_{k=1}^{K}\sum_{h=1}^{H}\mathbb{V}\left(P_{s_h^k,a_h^k,h},\widehat{V}_{h+1,\widetilde{r}}^{\pi^k}\right) \leq \widetilde{O}(H^2 K + \sqrt{H^5 K} + SAH^3).$$

$\square$

# B  Primal-dual Optimization Analysis

**Lemma 11.** *For the relaxed feasibility setting, let $b' = b + \tau$, for some $\tau > 0$, we have*
$$\widehat{\lambda}^{k,*} \leq \frac{H}{\tau}.$$
*For the strict feasibility setting, let $b' = b - \Delta$, for some $\Delta \in (0,\zeta)$, we have*
$$\widehat{\lambda}^{k,*} \leq \frac{H}{\zeta - \Delta}.$$

*Proof.* Writing the empirical CMDP in eq. (6) in its Lagrangian form,
$$\widehat{V}_{1,\widetilde{r}}^{\widehat{\pi}^{k,*}}(s_1^k) = \max_{\pi}\min_{\lambda \geq 0}\widehat{V}_{1,\widetilde{r}}^{\pi}(s_1^k) - \lambda\left(\widehat{V}_{1,\underline{c}}^{\pi}(s_1^k) - b'\right)$$

Using the linear programming formulation of CMDPs in terms of the state-occupancy measures $\mu$, we know that both the objective and the constraint are linear functions of $\mu$, and strong duality holds w.r.t. $\mu$. Since $\mu$ and $\pi$ have a one-to-one mapping, we can switch the min and the max, implying,
$$\widehat{V}_{1,\widetilde{r}}^{\widehat{\pi}^{k,*}}(s_1^k) = \min_{\lambda \geq 0}\max_{\pi}\widehat{V}_{1,\widetilde{r}}^{\pi}(s_1^k) - \lambda\left(\widehat{V}_{1,\underline{c}}^{\pi}(s_1^k) - b'\right)$$

Since $\widehat{\lambda}^{k,*}$ is the optimal dual variable for the empirical CMDP in eq. (6) and recall $\pi_c^* \in \arg\max_{\pi}b - V_{1,r}^{\pi}(s_1)$, we have
$$\widehat{V}_{1,\widetilde{r}}^{\widehat{\pi}^{k,*}}(s_1^k) = \max_{\pi}\widehat{V}_{1,\widetilde{r}}^{\pi}(s_1^k) - \widehat{\lambda}^{k,*}\left(\widehat{V}_{1,\underline{c}}^{\pi}(s_1^k) - b'\right)$$
$$\geq \widehat{V}_{1,\widetilde{r}}^{\pi_c^*}(s_1^k) - \widehat{\lambda}^{k,*}\left(\widehat{V}_{1,\underline{c}}^{\pi_c^*}(s_1^k) - b'\right)$$
$$= \widehat{V}_{1,\widetilde{r}}^{\pi_c^*}(s_1^k) + \widehat{\lambda}^{k,*}\left((b'-b) + (b - V_{1,c}^{\pi_c^*}(s_1^k)) + (V_{1,c}^{\pi_c^*}(s_1^k) - \widehat{V}_{1,\underline{c}}^{\pi_c^*}(s_1^k))\right)$$

Note that $\pi_c^*$ is a fixed policy and by lemma 16 we have $V_{1,c}^{\pi_c^*}(s_1^k) - \widehat{V}_{1,\underline{c}}^{\pi_c^*}(s_1^k) \geq 0$. Recall that $\zeta = b - V_{1,c}^{\pi_c^*}(s_1)$, and we have

$$\geq \widehat{V}_{1,\widetilde{r}}^{\pi_c^*}(s_1^k) + \widehat{\lambda}^{k,*}\left((b'-b) + \zeta\right).$$

For the relaxed feasibility setting, we let $b' = b + \tau$ for some $\tau > 0$. Then we have

$$\widehat{\lambda}^{k,*} \leq \frac{\widehat{V}_{1,\widetilde{r}}^{\widehat{\pi}^{k,*}}(s_1^k) - \widehat{V}_{1,\widetilde{r}}^{\pi_c^*}(s_1^k)}{\tau + \zeta} \leq \frac{H}{\tau}.$$

Note that the shift $\tau > 0$ is introduced to get rid of $\zeta$ in the results of the relaxed feasibility setting. For the strict feasibility setting, let $b' = b - \Delta$ for some $\Delta \in (0, \zeta)$, we have

$$\widehat{\lambda}^{k,*} \leq \frac{\widehat{V}_{1,\widetilde{r}}^{\widehat{\pi}^{k,*}}(s_1^k) - \widehat{V}_{1,\widetilde{r}}^{\pi_c^*}(s_1^k)}{-\Delta + \zeta} \leq \frac{H}{\zeta - \Delta}.$$

$\square$

**Lemma 12.** *Let $U > \widehat{\lambda}^{k,*}$ and for any $\widetilde{\pi}$ s.t.*

$$\widehat{V}_{1,\widetilde{r}}^{\widehat{\pi}^{k,*}}(s_1^k) - \widehat{V}_{1,\widetilde{r}}^{\widetilde{\pi}}(s_1^k) + U\left(\widehat{V}_{1,\underline{c}}^{\widetilde{\pi}}(s_1^k) - b'\right)_+ \leq B,$$

*we have*

$$\left(\widehat{V}_{1,\underline{c}}^{\widetilde{\pi}}(s_1^k) - b'\right)_+ \leq \frac{B}{U - \widehat{\lambda}^{k,*}}.$$

*Proof.* Define $\nu(\gamma) = \max_\pi \{\widehat{V}_{1,\widetilde{r}}^\pi(s_1^k) \mid \widehat{V}_{1,\underline{c}}^\pi(s_1^k) \leq b' - \gamma\}$ and note that by definition, $\nu(0) = \widehat{V}_{1,\widetilde{r}}^{\widehat{\pi}^{k,*}}(s_1^k)$, and that $\nu$ is a decreasing function for its argument. Then, for any policy $\pi$ s.t. $\widehat{V}_{1,\underline{c}}^\pi(s_1^k) \leq b' - \gamma$, we have

$$\begin{aligned}
\widehat{V}_{1,\widetilde{r}}^\pi(s_1^k) - \widehat{\lambda}^{k,*}(\widehat{V}_{1,\underline{c}}^\pi(s_1^k) - b') &\leq \max_\pi \widehat{V}_{1,\widetilde{r}}^\pi(s_1^k) - \widehat{\lambda}^{k,*}(\widehat{V}_{1,\underline{c}}^\pi(s_1^k) - b') \\
&= \widehat{V}_{1,\widetilde{r}}^{\widehat{\pi}^{k,*}}(s_1^k) - \widehat{\lambda}^{k,*}(\widehat{V}_{1,\underline{c}}^{\widehat{\pi}^{k,*}}(s_1^k) - b') \\
&= \widehat{V}_{1,\widetilde{r}}^{\widehat{\pi}^{k,*}}(s_1^k) = \nu(0) \quad \text{(by strong duality)}
\end{aligned}$$

This further implies

$$\begin{aligned}
\nu(0) - \widehat{\lambda}^{k,*}\gamma &\geq \widehat{V}_{1,\widetilde{r}}^\pi(s_1^k) - \widehat{\lambda}^{k,*}(\widehat{V}_{1,\underline{c}}^\pi(s_1^k) - b') - \widehat{\lambda}^{k,*}\gamma \\
&= \widehat{V}_{1,\widetilde{r}}^\pi(s_1^k) - \widehat{\lambda}^{k,*}(\widehat{V}_{1,\underline{c}}^\pi(s_1^k) - (b' - \gamma))
\end{aligned}$$

Since this holds for any policy $\pi$ s.t. $\widehat{V}_{1,\underline{c}}^\pi(s_1^k) \leq b' - \gamma$, we have

$$\nu(0) - \widehat{\lambda}^{k,*}\gamma \geq \max_\pi \{\widehat{V}_{1,\widetilde{r}}^\pi(s_1^k) \mid \widehat{V}_{1,\underline{c}}^\pi(s_1^k) \leq b' - \gamma\} = \nu(\gamma),$$

and thus

$$\widehat{\lambda}^{k,*}\gamma \leq \nu(0) - \nu(\gamma).$$

Now we choose $\widetilde{\gamma} = -(\widehat{V}_{1,\underline{c}}^{\widetilde{\pi}}(s_1^k) - b')_+$,

$$\begin{aligned}
U - \widehat{\lambda}^{k,*}|\widetilde{\gamma}| = \widehat{\lambda}^{k,*}\widetilde{\gamma} + U|\widetilde{\gamma}| &\leq \nu(0) - \nu(\widetilde{\gamma}) + U|\widetilde{\gamma}| \\
&= \widehat{V}_{1,\widetilde{r}}^{\widehat{\pi}^{k,*}}(s_1^k) - \widehat{V}_{1,\widetilde{r}}^{\widetilde{\pi}}(s_1^k) + U|\widetilde{\gamma}| + \widehat{V}_{1,\widetilde{r}}^{\widetilde{\pi}}(s_1^k) - \nu(\widetilde{\gamma}) \\
&= \widehat{V}_{1,\widetilde{r}}^{\widehat{\pi}^{k,*}}(s_1^k) - \widehat{V}_{1,\widetilde{r}}^{\widetilde{\pi}}(s_1^k) + U(\widehat{V}_{1,\underline{c}}^{\widetilde{\pi}}(s_1^k) - b')_+ + \widehat{V}_{1,\widetilde{r}}^{\widetilde{\pi}}(s_1^k) - \nu(\widetilde{\gamma}) \\
&\leq B + \widehat{V}_{1,\widetilde{r}}^{\widetilde{\pi}}(s_1^k) - \nu(\widetilde{\gamma}).
\end{aligned}$$

Now let us bound $\nu(\widetilde{\gamma})$:

$$\nu(\widetilde{\gamma}) = \max_{\pi}\{\widehat{V}^{\pi}_{1,\widetilde{r}}(s^k_1) \mid \widehat{V}^{\pi}_{1,\underline{c}}(s^k_1) \le b' + (\widehat{V}^{\widetilde{\pi}}_{1,\underline{c}}(s^k_1) - b')_+\}$$

$$\ge \max_{\pi}\{\widehat{V}^{\pi}_{1,\widetilde{r}}(s^k_1) \mid \widehat{V}^{\pi}_{1,\underline{c}}(s^k_1) \le \widehat{V}^{\widetilde{\pi}}_{1,\underline{c}}(s^k_1)\} \quad \text{(tightening the constraint)}$$

$$\ge \widehat{V}^{\widetilde{\pi}}_{1,\widetilde{r}}(s^k_1).$$

Finally,

$$U - \widehat{\lambda}^{k,*}|\widetilde{\gamma}| \le B \implies (\widehat{V}^{\widetilde{\pi}}_{1,\underline{c}}(s^k_1) - b')_+ \le \frac{B}{U - \widehat{\lambda}^{k,*}}.$$

$\square$

**Lemma 13.**

$$\widehat{V}^{\widehat{\pi}^{k,*}}_{1,\widetilde{r}}(s^k_1) - \widehat{V}^{\pi^k}_{1,\widetilde{r}}(s^k_1) + \lambda\left(\widehat{V}^{\pi^k}_{1,\underline{c}}(s^k_1) - b'\right) \le \frac{1}{T}\sum_{t=1}^{T}(\widehat{\lambda}^k_t - \lambda)\left(b' - \widehat{V}^{\widehat{\pi}^k_t}_{1,\underline{c}}(s^k_1)\right).$$

*Proof.* For any episode $k$ and any time step $t$ in the primal-dual iterations, the primal update ensures that for any policy $\pi$,

$$\widehat{V}^{\widehat{\pi}^k_t}_{1,\widetilde{r}}(s^k_1) - \widehat{\lambda}^k_t(\widehat{V}^{\widehat{\pi}^k_t}_{1,\underline{c}}(s^k_1) - b') \ge \widehat{V}^{\pi}_{1,\widetilde{r}}(s^k_1) - \widehat{\lambda}^k_t(\widehat{V}^{\pi}_{1,\underline{c}}(s^k_1) - b').$$

Let $\pi$ be $\widehat{\pi}^{k,*}$, and rearrange:

$$\widehat{V}^{\widehat{\pi}^{k,*}}_{1,\widetilde{r}}(s^k_1) - \widehat{V}^{\widehat{\pi}^k_t}_{1,\widetilde{r}}(s^k_1) \le \widehat{\lambda}^k_t(\widehat{V}^{\widehat{\pi}^{k,*}}_{1,\underline{c}}(s^k_1) - \widehat{V}^{\widehat{\pi}^k_t}_{1,\underline{c}}(s^k_1)).$$

Note that $\widehat{\pi}^{k,*}$ is the solution to the empirical CMDP in eq. (6), thus $\widehat{V}^{\widehat{\pi}^{k,*}}_{1,\underline{c}}(s^k_1) \le b'$, and we have

$$\widehat{V}^{\widehat{\pi}^{k,*}}_{1,\widetilde{r}}(s^k_1) - \widehat{V}^{\widehat{\pi}^k_t}_{1,\widetilde{r}}(s^k_1) \le \widehat{\lambda}^k_t(b' - \widehat{V}^{\widehat{\pi}^k_t}_{1,\underline{c}}(s^k_1)).$$

Take average over $T$ iterations,

$$\frac{1}{T}\sum_{t=1}^{T}\left(\widehat{V}^{\widehat{\pi}^{k,*}}_{1,\widetilde{r}}(s^k_1) - \widehat{V}^{\widehat{\pi}^k_t}_{1,\widetilde{r}}(s^k_1)\right) \le \frac{1}{T}\sum_{t=1}^{T}\widehat{\lambda}^k_t\left(b' - \widehat{V}^{\widehat{\pi}^k_t}_{1,\underline{c}}(s^k_1)\right).$$

To use lemma 14, we rewrite as

$$\frac{1}{T}\sum_{t=1}^{T}\left(\widehat{V}^{\widehat{\pi}^{k,*}}_{1,\widetilde{r}}(s^k_1) - \widehat{V}^{\widehat{\pi}^k_t}_{1,\widetilde{r}}(s^k_1)\right) + \frac{1}{T}\sum_{t=1}^{T}\lambda\left(\widehat{V}^{\widehat{\pi}^k_t}_{1,\underline{c}}(s^k_1) - b'\right) \le \frac{1}{T}\sum_{t=1}^{T}(\widehat{\lambda}^k_t - \lambda)\left(b' - \widehat{V}^{\widehat{\pi}^k_t}_{1,\underline{c}}(s^k_1)\right).$$

Note that $\widehat{V}^{\widehat{\pi}^{k,*}}_{1,\widetilde{r}}(s^k_1)$ is constant throughout $T$ primal-dual iterations, and $\pi^k$ is a mixture policy, then

$$\widehat{V}^{\widehat{\pi}^{k,*}}_{1,\widetilde{r}}(s^k_1) - \widehat{V}^{\pi^k}_{1,\widetilde{r}}(s^k_1) + \lambda\left(\widehat{V}^{\pi^k}_{1,\underline{c}}(s^k_1) - b'\right) \le \frac{1}{T}\sum_{t=1}^{T}(\widehat{\lambda}^k_t - \lambda)\left(b' - \widehat{V}^{\widehat{\pi}^k_t}_{1,\underline{c}}(s^k_1)\right).$$

$\square$

**Lemma 14.** *Recall we set $\eta = \frac{U}{H\sqrt{T}}$. For any episode $k$, any $\lambda \in [0, U]$, and primal and dual updates in eqs. (8) and (9), we have*

$$\frac{1}{T}\sum_{t=1}^{T}\left(\widehat{\lambda}^k_t - \lambda\right)\left(b' - \widehat{V}^{\widehat{\pi}^k_t}_{1,\underline{c}}(s^k_1)\right) \le 2\varepsilon_1 H\sqrt{T} + \frac{UH}{\sqrt{T}}.$$

*Proof.* In this proof, for the simplicity of notations, we will only focus on primal-dual iterations in an arbitrary episode $k \in [K]$, and thus we will drop all dependency on $k$ when the context is clear. The dual update is given by

$$\widehat{\lambda}_{t+1} = \mathcal{R}_\Lambda[\widehat{\lambda}_t - \eta(b' - \widehat{V}^{\widehat{\pi}_t}_{1,\underline{c}}(s_1))].$$

Particularly, we denote

$$\widehat{\lambda}'_{t+1} = P_{[0,U]}[\widehat{\lambda}_t - \eta(b' - \widehat{V}^{\widehat{\pi}_t}_{1,\underline{c}}(s_1))].$$

First, we shall look at $|\widehat{\lambda}_t - \lambda|$:

$$
\begin{aligned}
|\widehat{\lambda}_{t+1} - \lambda| = |\mathcal{R}_\Lambda[\widehat{\lambda}'_{t+1}] - \lambda| &= |\mathcal{R}_\Lambda[\widehat{\lambda}'_{t+1}] - \widehat{\lambda}'_{t+1} + \widehat{\lambda}'_{t+1} - \lambda| \\
&\leq |\mathcal{R}_\Lambda[\widehat{\lambda}'_{t+1}] - \widehat{\lambda}'_{t+1}| + |\widehat{\lambda}'_{t+1} - \lambda| \\
&\leq \varepsilon_1 + |\widehat{\lambda}'_{t+1} - \lambda|.
\end{aligned}
$$

Take square on both sides,

$$
\begin{aligned}
|\widehat{\lambda}_{t+1} - \lambda|^2 &\leq \varepsilon_1^2 + 2\varepsilon_1|\widehat{\lambda}'_{t+1} - \lambda| + |\widehat{\lambda}'_{t+1} - \lambda|^2 \\
&\leq \varepsilon_1^2 + 2\varepsilon_1 U + |\widehat{\lambda}'_{t+1} - \lambda|^2 \\
&\leq \varepsilon_1^2 + 2\varepsilon_1 U + |\widehat{\lambda}_t - \eta(b' - \widehat{V}^{\widehat{\pi}_t}_{1,\underline{c}}(s_1)) - \lambda|^2 \\
&= \varepsilon_1^2 + 2\varepsilon_1 U + |\widehat{\lambda}_t - \lambda|^2 - 2\eta(b' - \widehat{V}^{\widehat{\pi}_t}_{1,\underline{c}}(s_1))(\widehat{\lambda}_t - \lambda) + \eta^2(b' - \widehat{V}^{\widehat{\pi}_t}_{1,\underline{c}}(s_1))^2 \\
&\leq \varepsilon_1^2 + 2\varepsilon_1 U + |\widehat{\lambda}_t - \lambda|^2 - 2\eta(b' - \widehat{V}^{\widehat{\pi}_t}_{1,\underline{c}}(s_1))(\widehat{\lambda}_t - \lambda) + \eta^2 H^2.
\end{aligned}
$$

Now we have

$$(\widehat{\lambda}_t - \lambda)(b' - \widehat{V}^{\widehat{\pi}_t}_{1,\underline{c}}(s_1)) \leq \frac{\varepsilon_1^2 + 2\varepsilon_1 U + \eta^2 H^2}{2\eta} + \frac{|\widehat{\lambda}_t - \lambda|^2 - |\widehat{\lambda}_{t+1} - \lambda|^2}{2\eta}.$$

By taking average over $T$ iterations and telescoping, we have

$$
\begin{aligned}
\frac{1}{T}\sum_{t=1}^{T}(\widehat{\lambda}_t - \lambda)(b' - \widehat{V}^{\widehat{\pi}_t}_{1,\underline{c}}(s_1)) &\leq \frac{\varepsilon_1^2 + 2\varepsilon_1 U + \eta^2 H^2}{2\eta} + \frac{|\lambda_1 - \lambda|^2 - |\lambda_{T+1} - \lambda|^2}{2\eta T} \\
&\leq \frac{\varepsilon_1^2 + 2\varepsilon_1 U + \eta^2 H^2}{2\eta} + \frac{|\lambda_1 - \lambda|^2}{2\eta T} \\
&\leq \frac{\varepsilon_1^2 + 2\varepsilon_1 U + \eta^2 H^2}{2\eta} + \frac{U^2}{2\eta T} \\
&\leq \frac{2\varepsilon_1 U}{\eta} + \frac{\eta H^2}{2} + \frac{U^2}{2\eta T} \\
&= 2\varepsilon_1 H\sqrt{T} + \frac{UH}{\sqrt{T}}.
\end{aligned}
$$

$\square$

## C  Useful Lemmas

**Lemma 15** (Primal update)**.** *We re-state the primal update in eq.* (8)*:*

$$\widehat{\pi}^k_t = \arg\max_{\pi} \widehat{V}^\pi_{1,\widetilde{r}}(s_1) - \widehat{\lambda}^k_t\left(\widehat{V}^\pi_{1,\underline{c}}(s_1) - b'\right) = \arg\max_{\pi} \widehat{V}^\pi_{1,\widetilde{r}-\widehat{\lambda}^k_t\underline{c}}(s_1).$$

*Proof.* Note the estimate value functions $\widehat{V}^\pi_{1,g}$ of any reward-like function $g : \mathcal{S} \times \mathcal{A} \to \mathbb{R}$ is given by:

$$\widehat{V}^\pi_{1,g}(s) = \mathbb{E}_{\widehat{P},\pi}\left[\sum_{t=1}^{H} g(S_t, A_t)|S_1 = s\right].$$

Then we have

$$\widehat{\pi}_t^k = \arg\max_{\pi} \widehat{V}_{1,\widetilde{r}}^{\pi}(s_1) - \widehat{\lambda}_t^k \left( \widehat{V}_{1,\underline{c}}^{\pi}(s_1) - b' \right)$$

$$= \arg\max_{\pi} \mathbb{E}_{\widehat{P},\pi} \left[ \sum_{t=1}^{H} \widetilde{r}(S_t, A_t) | S_1 = s_1 \right] - \widehat{\lambda}_t^k \cdot \mathbb{E}_{\widehat{P},\pi} \left[ \sum_{t=1}^{H} \underline{c}(S_t, A_t) | S_1 = s_1 \right] + \widehat{\lambda}_t^k \cdot b'.$$

$$= \arg\max_{\pi} \mathbb{E}_{\widehat{P},\pi} \left[ \sum_{t=1}^{H} (\widetilde{r} - \widehat{\lambda}_t^k \underline{c})(S_t, A_t) | S_1 = s_1 \right]$$

$$= \arg\max_{\pi} \widehat{V}_{1,\widetilde{r}-\widehat{\lambda}_t^k \underline{c}}^{\pi}(s_1),$$

where the second-to-last equality follows from the linearity of expectation and reward-like functions, and the last equality follows from the definition of estimate value functions. $\square$

**Lemma 16** (Optimism). *With probability at least $1 - 2\delta'$, for any fixed policy $\pi$, reward function $g$ and $s \in \mathcal{S}, h \in [H]$, we have*

$$\widehat{V}_{h,\widetilde{g}}^{\pi}(s) \geq V_{h,g}^{\pi}(s) \geq \widehat{V}_{h,\underline{g}}^{\pi}(s).$$

*Proof.* First, we define the following function

$$f(p, v, n) := \langle p, v \rangle + \max \left\{ \frac{20}{3} \sqrt{\frac{\mathbb{V}(p,v) \log \frac{1}{\delta'}}{n}}, \frac{400}{9} \frac{H \log \frac{1}{\delta'}}{n} \right\}$$

for any vector $p \in \Delta^S$, any non-negative vector $v \in \mathbb{R}^S$ obeying $\|v\|_\infty \leq H$, and any positive integer $n$. We claim that

$$f(p, v, n) \text{ is non-decreasing in each entry of } v. \tag{17}$$

To justify this claim, consider any $1 \leq s \leq S$, and let us freeze $p, n$ and all but the $s$-th entries of $v$. It then suffices to observe that (i) $f$ is a continuous function, and (ii) except for at most two possible choices of $v(s)$ that obey $\frac{20}{3}\sqrt{\frac{V(p,v)\log\frac{1}{\delta'}}{n}} = \frac{400}{9}\frac{H\log\frac{1}{\delta'}}{n}$, one can use the properties of $p$ and $v$ to calculate

$$\frac{\partial f(p,v,n)}{\partial v(s)} = p(s) + \frac{20}{3}\mathbb{1}\left\{ \frac{20}{3}\sqrt{\frac{\mathbb{V}(p,v)\log\frac{1}{\delta'}}{n}} \geq \frac{400}{9}\frac{H\log\frac{1}{\delta'}}{n} \right\} \frac{p(s)(v(s) - \langle p, v \rangle)\sqrt{\log\frac{1}{\delta'}}}{\sqrt{n\mathbb{V}(p,v)}}$$

$$= p(s) + \mathbb{1}\left\{ \sqrt{n\mathbb{V}(p,v)\log\frac{1}{\delta'}} \geq \frac{20}{3}H\log\frac{1}{\delta'} \right\} \frac{\frac{20}{3}H\log\frac{1}{\delta'}}{\sqrt{n\mathbb{V}(p,v)\log\frac{1}{\delta'}}} \cdot \frac{p(s)(v(s) - \langle p, v \rangle)}{H}$$

$$\geq \min\left\{ p(s) + p(s)\frac{(v(s) - \langle p, v \rangle)}{H}, p(s) \right\}$$

$$\geq p(s)\min\left\{ \frac{H + v(s) - \langle p, v \rangle}{H}, 1 \right\} \geq 0,$$

thus establishing the claim. We now proceed to the proof of lemma 16. Consider any $(h, k, s, a)$, and we divide into two cases.

Case 1: $N_h^k(s,a) \leq 2$. In this case, the following trivial bounds arise directly from the value function initiation:

$$\widehat{Q}_{h,\widetilde{g}}^{\pi}(s,a) = H \geq Q_{h,g}^{\pi}(s,a) \geq 0 = \widehat{Q}_{h,\underline{g}}^{\pi}(s,a),$$

$$\widehat{V}_{h,\widetilde{g}}^{\pi}(s) = H \geq V_{h,g}^{\pi}(s) \geq 0 = \widehat{V}_{h,\underline{g}}^{\pi}(s).$$

Case 2: $N_h^k(s,a) > 2$. Suppose now that $\widehat{Q}_{h+1,\widetilde{g}}^{\pi} \geq Q_{h+1,g}^{\pi} \geq \widehat{Q}_{h+1,\underline{g}}^{\pi}$, which also implies that $\widehat{V}_{h+1,\widetilde{g}}^{\pi} \geq V_{h+1,g}^{\pi} \geq \widehat{V}_{h+1,\underline{g}}^{\pi}$. If $\widehat{Q}_{h,\widetilde{g}}^{\pi}(s,a) = H$, then $\widehat{Q}_{h,\widetilde{g}}^{\pi}(s,a) \geq Q_{h,g}^{\pi}(s,a)$ holds trivially, and

hence it suffices to look at the case with $\widehat{Q}^\pi_{h,\widetilde{g}}(s,a) < H$. According to the update rule, it holds that

$$\widehat{Q}^\pi_{h,\widetilde{g}}(s,a)$$

$$= g_h(s,a) + \left\langle \widehat{P}_{s,a,h}, \widehat{V}^\pi_{h+1,\widetilde{g}} \right\rangle + c_1 \sqrt{\frac{\mathbb{V}\left(\widehat{P}^k_{s,a,h}, \widehat{V}^\pi_{h+1,\widetilde{g}}\right) \log \frac{1}{\delta'}}{N^k_h(s,a)}} + c_2 \frac{H \log \frac{1}{\delta'}}{N^k_h(s,a)}$$

$$\geq g_h(s,a) + \frac{48H \log \frac{1}{\delta'}}{3N^k_h(s,a)} + f\left(\widehat{P}^k_{s,a,h}, \widehat{V}^\pi_{h+1,\widetilde{g}}, N^k_h(s,a)\right)$$

$$\geq g_h(s,a) + \frac{48H \log \frac{1}{\delta'}}{3N^k_h(s,a)} + f\left(\widehat{P}^k_{s,a,h}, V^\pi_{h+1,g}, N^k_h(s,a)\right)$$

for any $(s,a)$, where the last inequality results from the claim (eq. (17)) and the hypothesis $\widehat{V}^\pi_{h+1,\widetilde{g}} \geq V^\pi_{h+1,g}$. Moreover, applying Lemma 19, we have

$$\mathbb{P}\left\{\left|\left\langle \widehat{P}^k_{s,a,h} - P_{s,a,h}, V^\pi_{h+1,g}\right\rangle\right| > 2\sqrt{\frac{\mathbb{V}\left(\widehat{P}^k_{s,a,h}, V^\pi_{h+1,g}\right) \log \frac{1}{\delta'}}{N^k_h(s,a)}} + \frac{14H \log \frac{1}{\delta'}}{3N^k_h(s,a)}\right\}$$

$$\leq \mathbb{P}\left\{\left|\left\langle \widehat{P}^k_{s,a,h} - P_{s,a,h}, V^\pi_{h+1,g}\right\rangle\right| > \sqrt{\frac{2\mathbb{V}\left(\widehat{P}^k_{s,a,h}, V^\pi_{h+1,g}\right) \log \frac{1}{\delta'}}{N^k_h(s,a) - 1}} + \frac{7H \log \frac{1}{\delta'}}{3N^k_h(s,a) - 1}\right\} \leq 2\delta'.$$

This implies that with probability at least $1 - 2\delta'$,

$$f\left(\widehat{P}^k_{s,a,h}, V^\pi_{h+1,g}, N^k_h(s,a)\right) = \left\langle P_{s,a,h}, V^\pi_{h+1,g}\right\rangle + \left\langle \widehat{P}^k_{s,a,h} - P_{s,a,h}, V^\pi_{h+1,g}\right\rangle$$

$$+ \max\left\{\frac{20}{3}\sqrt{\frac{\mathbb{V}\left(\widehat{P}^k_{s,a,h}, V^\pi_{h+1,g}\right) \log \frac{1}{\delta'}}{N^k_h(s,a)}}, \frac{400}{9}\frac{H \log \frac{1}{\delta'}}{N^k_h(s,a)}\right\}$$

$$\geq \left\langle P_{s,a,h}, V^\pi_{h+1,g}\right\rangle.$$

Substitution into eq. (18) gives: with probability at least $1 - 2\delta'$,

$$\widehat{Q}^\pi_{h,\widetilde{g}}(s,a) \geq g_h(s,a) + \left\langle P_{s,a,h}, V^\pi_{h+1,g}\right\rangle = Q^\pi_{h,g}(s,a).$$

The proof for $Q^\pi_{h,g} \geq \widehat{Q}^\pi_{h,\underline{g}}$ is analogous and we leave out here. $\qquad\square$

**Lemma 17.** *Recall the definition of $N^k_h(s^k_h, a^k_h)$ in algorithm 1. It holds that:*

$$\sum_{k=1}^K \sum_{h=1}^H \frac{1}{\max\{N^k_h(s^k_h, a^k_h), 1\}} \leq 2SAH \log_2 K.$$

*Proof.* In view of the doubling batch update rule, it is easily seen that: for any given $(s,a,h)$,

$$\sum_{k=1}^K \frac{1}{\max\{N^k_h(s^k_h, a^k_h), 1\}} \mathbb{1}\left\{(s,a) = (s^k_h, a^k_h)\right\} \leq 2\log_2 K,$$

since each $(s,a,h)$ is associated with at most $\log_2 K$ epochs. Summing over $(s,a,h)$ completes the proof. $\qquad\square$

**Lemma 18** (Freedmans inequality)**.** *Let $(M_n)_{n\geq 0}$ be a martingale such that $M_0 = 0$ and $|M_n - M_{n-1}| \leq c \ (\forall n \geq 1)$ hold for some quantity $c > 0$. Define*

$$Var_n := \sum_{k=1}^n \mathbb{E}\left[(M_k - M_{k-1})^2 \big| \mathcal{F}_{k-1}\right]$$

*for every $n \geq 0$, where $\mathcal{F}_k$ is the $\sigma$-algebra generated by $(M_1, \ldots, M_k)$. Then for any integer $n \geq 1$ and any $\epsilon, \delta > 0$, one has*

$$\mathbb{P}\left[|M_n| \geq 2\sqrt{2}\sqrt{Var_n \log \frac{1}{\delta}} + 2\sqrt{\epsilon \log \frac{1}{\delta}} + 2c \log \frac{1}{\delta}\right] \leq 2\left(\log_2\left(\frac{nc^2}{\epsilon}\right) + 1\right)\delta.$$

