# OpenReview forum: "Near-Optimal Sample Complexity for Online Constrained MDPs"
_NeurIPS.cc/2025/Conference — NeurIPS 2025 poster_

### Official Review · Reviewer_JXZe · 2025-06-08

**Clarity:** 3
**Significance:** 2
**Originality:** 3
**Rating:** 4
**Confidence:** 4

**Summary:**

The paper studies tabular CMDPs. They proposed a primal-dual algorithm that obtains a near-optimal policy with sample complexity that matches the lower bound. This holds for both the setting considered in the paper, relaxed and strict feasibility.
The paper is generally well-written and easy to follow. I had several questions while reading—some were answered later, but it would still be helpful to clarify them in the final version for the benefit of other readers.

**Questions:**

What does giving access to generative model means? Does it imply that system can obtain transitions values for any state, looks impractical?

line 71-71: It was counterintuitive why sample complexity depends inversely on the size of the feasible region.

line 162: explain intutively \tilde{r}. Is it some optimistic acquisition function?

In section 3: problem setup can you write the finite state space and finite action spaces? It is mentioned in line 36 but no where in problem setup section.

Methodology section starts nicely with explaining why the existing methods have loose bounds. Since the section is long with a sequence of paragraphs, i recommend naming paragraphs as "\textbf{name.} content of paragraph....."

line 174-181: Can you also clarify why the doubling batch updates solves the problem

It would be good to mention t, k, and h early on and what they represent. In line 202 why does a finite number of solve iterations T is Guaranteed to converge to optimal solution?

line 233, why \pi^{\tu, \star} is a deterministic policy?

in the proof of theorem 1, why does s_1 depends on k? is it a typo? If it does depend on k, then \pi^{\tau, \star} should also depend on k.

line 237-238: Is this term (3rd regret term) non-zero due to finite number of gradient steps T? If it T is long enough (converges) then it will be zero as well? I think so, but it is not evident with the upper bounds of this term. Is your regret optimal with T?

What does _{+} represents is CV(K)? If it is ciel function, shouldn't it be inside \sum_k ?

Theorem 4: why there is constraint violations in the case of strict feasibility? (c.f. line 14 saying zero violation)

theorem 3: does \delta' h in line line 264 is same as line 184? Since theorem statement is with 1-\delta, may be just write 1-SHAK\delta' > 1-\delta.

**Ethical Concerns:**

["NO or VERY MINOR ethics concerns only"]

**Final Justification:**

I would keep the score as borderline accept. The paper makes a theoretical contribution by providing upper bounds that match existing lower bounds. However, the considered setting of tabular, episodic MDP is very limiting, and the absence of a simulation or numerical study leads me to keep it borderline.

**Limitations:**

yes

**Quality:**

3

**Strengths And Weaknesses:**

Strength:
The theoretical results are solid. They provide upper bounds for their algorithm, which match the lower bounds existing in the literature.

Weakness: There were no major flaws/weaknesses in the paper.
- The assumption of a finite state-action space is somewhat limiting in practice. The model essentially reduces to a linear setting. (This is only my opinion about the considered settings in the literature; however, there are multiple works with similar setting,s and this paper adds value)
- The paper would benefit from an implementation of the algorithm and releasing the code.
- It would be helpful to clearly state all assumptions using an Assumption environment and refer to them explicitly in the theorems. (But not sure if there are any assumptions apart from written in the problem setup) For example, do you require full support over the state space from the initial state distribution?
- See the questions below for improving paper clarity

---

> ### Author Rebuttal · Authors · 2025-07-31
>
> We thank the reviewer for the constructive feedback, insightful comments, and positive assessment of our work.
>
> We address the reviewer's concerns below:
> - *The assumption of a finite state-action space is somewhat limiting in practice. The model essentially reduces to a linear setting. (This is only my opinion about the considered settings in the literature; however, there are multiple works with similar setting,s and this paper adds value)*
>
> Tabular CMDPs provide the fundamental setting for more complex settings such as function approximations, and it is of significance to obtain minimax optimal results for online tabular CMDPs. We agree that generalizing our techniques and results is important and view it as an exciting direction for future work.
>
> - *The paper would benefit from an implementation of the algorithm and releasing the code.*
>
> We agree that empirical evaluation of CMDP algorithms is valuable. Our primary contribution of this paper is theoretical and to close the gap between upper and lower bounds for online tabular CMDPs using novel analysis techniques. While comprehensive empirical study is beyond the scope of this paper, we plan to conduct numerical experiments in future work.
>
> - *It would be helpful to clearly state all assumptions using an Assumption environment and refer to them explicitly in the theorems. (But not sure if there are any assumptions apart from written in the problem setup) For example, do you require full support over the state space from the initial state distribution?*
>
> We thank the reviewer for the constructive suggestion. We will highlight the known reward and cost functions assumption in the final version. We do not require full support over the state space from the initial state distribution.
>
> ---
>
> We respond to the reviewer's questions below:
> - *What does giving access to generative model means? Does it imply that system can obtain transitions values for any state, looks impractical?*
>
> A generative model, also called random access, gives the learner oracle access to a simulator that, for any queried $(s,a,h)$ tuple, returns an i.i.d. sample of the next state, reward, and cost drawn from the true distributions. A generative model is indeed rarely available in practice. Therefore, extending minimax optimal results from the random access setting to the online learning setting is significant in the CMDP literature.
>
> - *line 71-71: It was counterintuitive why sample complexity depends inversely on the size of the feasible region.*
>
> Intuitively, the Slater constant $\zeta$ measures the slack in the constraint. For a small $\zeta$, a small estimation error can make an otherwise feasible policy appear infeasible. To guarantee the same reward suboptimality, the learner must estimate reward and cost values with higher precision, demanding a higher sample complexity.
>
> - *line 162: explain intuitively $\tilde{r}$. Is it some optimistic acquisition function?*
>
> The optimistic reward $\tilde{r} = r + b_r$ and optimistic cost $\underline{c} = c - b_c$ is introduced purely for notational clarity: the UCB bonus $b_h(s,a)$, which compensates for transition uncertainty, is absorbed into the reward. Writing value estimates (e.g., $\hat{V}\_{h,\tilde{r}}(s)$ and $\hat{V}\_{h, \tilde{r} - \lambda\underline{c}}(s)$) with $\tilde{r}$ and $\underline{c}$ makes it explicit how bonus terms enter each estimate, keeping the exposition concise.
>
> - *In section 3: problem setup can you write the finite state space and finite action spaces? It is mentioned in line 36 but no where in problem setup section.*
>
> We appreciate the suggestion. In the camera-ready version, section 3 will explicitly state that both the state and action spaces are finite when we introduce the CMDP formalities.
>
> - *Methodology section starts nicely with explaining why the existing methods have loose bounds. Since the section is long with a sequence of paragraphs, i recommend naming paragraphs as "\textbf{name.} content of paragraph....."*
>
> We thank the reviewer for this constructive suggestion. We will add paragraph headings to better guide the readers in the final version.
>
> - *line 174-181: Can you also clarify why the doubling batch updates solves the problem*
>
> As mentioned in lines 167-170, although fixing a profile decouples the dependence between $\hat{P}\_h$ and $\hat{V}\_{h+1}$, it is the doubling batch updates that reduces the number of profiles, so a union bound over all profiles yields tight results.
>
> - *It would be good to mention $t$, $k$, and $h$ early on and what they represent. In line 202 why does a finite number of solve iterations $T$ is Guaranteed to converge to optimal solution?*
>
> We thank the reviewer for the helpful suggestion. We will revise the presentation in the final version to clarify that we do not require solving equation 7 exactly. Accordingly, we will update the wording from "solve the saddle-point problem eq. 7 iteratively" to "tackle the saddle-point problem eq.7 iteratively".
>
> - *line 233, why $\pi^{\tau, \*}$ is a deterministic policy?*
>
> We thank the reviewer for catching this. The policy $\pi^{\tau,\*}$ is in fact fixed, but not necessarily deterministic. We will revise the wording to "fixed" in line 233 to reflect that it may be randomized. Meanwhile, the policy $\pi^{\tau,\*}$ remains independent of the online learning process, and this typo does not affect the soundness of our theoretical results.
>
> - *in the proof of theorem 1, why does $s_1$ depends on $k$? is it a typo? If it does depend on $k$, then $\pi^{\tau, \*}$ should also depend on $k$.*
>
> We thank the reviewer for catching this typo. It should be $s_1$ throughout the paper. We will revise the notations in the final version.
>
> - *line 237-238: Is this term (3rd regret term) non-zero due to finite number of gradient steps $T$? If it $T$ is long enough (converges) then it will be zero as well? I think so, but it is not evident with the upper bounds of this term. Is your regret optimal with $T$?*
>
> The third term is non-zero due to two factors (c.f. lemma 14, second-to-last line of the last equation): (1) We discretize $\lambda$ by an $\varepsilon_1$-net. This introduces a non-zero optimization error of $\frac{2\varepsilon_1 U}{\eta}$, which depends on $\varepsilon_1$ and does not vanish as $T$ goes to infinity. (2) The $T$-iteration primal-dual optimization introduces an error of $\frac{\eta H^2}{2} + \frac{U^2}{2\eta T} = \frac{UH}{\sqrt{T}}$ (by selecting $\eta = \frac{U}{H\sqrt{T}}$), which indeed goes to zero as $T$ goes to infinity. Therefore, our results are optimal in $T$, as increasing $T$ will not reduce the leading terms.
>
> - *What does \_{+} represents is CV(K)? If it is ceil function, shouldn't it be inside sum\_k?*
>
> We will add the notation of $(x)_+ = \max\\{x, 0\\}$ to improve clarity. The cumulative constraint violation defined in equation 2 is used to compute the constraint violation of the output mixture policy in theorem 5 and 6. It is also worth mentioning that this definition of constraint violation is widely adopted in many CMDP works [1,2,3,4].
>
> - *Theorem 4: why there is constraint violations in the case of strict feasibility? (c.f. line 14 saying zero violation)*
>
> As mentioned in the definition of the strict feasibility regime (c.f. line 144-146), our algorithm returns a zero-violation policy. Moreover, with parameters defined in theorem 6, we can bound $CV(K) \le 0$.
>
> - *theorem 3: does $\delta'$ in line line 264 is same as line 184? Since theorem statement is with $1-\delta$, may be just write $1-SHAK\delta' \rightarrow 1-\delta$.*
>
> We thank the reviewer for the constructive comment. We will revise the probability guarantees in the final version to ensure consistency and clarity in the use of $\delta'$ and $\delta$.
>
> ---
>
> Reference
>
> [1] Dongsheng Ding, Xiaohan Wei, Zhuoran Yang, Zhaoran Wang, and Mihailo R. Jovanović. Provably efficient safe exploration via primal-dual policy optimization. In International Conference on Artificial Intelligence and Statistics, 2020.
>
> [2] Arushi Jain, Sharan Vaswani, Reza Babanezhad Harikandeh, Csaba Szepesvári, and Doina Precup. Towards painless policy optimization for constrained MDPs. In The 38th Conference on Uncertainty in Artificial Intelligence, 2022.
>
> [3] Tao Liu, Ruida Zhou, Dileep Kalathil, P. R. Kumar, and Chao Tian. Learning policies with zero or bounded constraint violation for constrained mdps. In Proceedings of the 35th International Conference on Neural Information Processing Systems, NIPS ’21, 2021.
>
> [4] Yonathan Efroni, Shie Mannor, and Matteo Pirotta. Exploration-exploitation in constrained mdps, 2020.

---

> > ### Comment · Reviewer_JXZe · 2025-08-01
> >
> > Thanks for the response. I still didn't understand
> >
> > - In line 202 why does a finite number of solve iterations $T$ is Guaranteed to converge to optimal solution?
> >
> > - Theorem 4: why there is constraint violations in the case of strict feasibility? (c.f. line 14 saying zero violation)

---

> > > ### Author Response · Authors · 2025-08-01
> > >
> > > - *In line 202 why does a finite number of solve iterations $T$ is Guaranteed to converge to optimal solution?*
> > >
> > > We would like to clarify that the exact convergence is neither guaranteed by the finite $T$ iterations of primal-dual updates, nor required by our algorithm: (1) The optimization error in the iterative primal-dual updates is inherent, regardless of the number of iterations $T$. As shown in lemma 2 and 14 and also discussed in our answer to the reviewer's Question 10 (*is the 3rd regret term non-zero due to a finite $T$?*), the discretization of $\lambda$ contributes to a $2\varepsilon\_1 U/\eta$ error, which does not vanish even if $T$ is set to be infinitely large. (2) Our algorithm does not require the exact convergence to the optimal solution. As shown in the proofs to theorem 1 and 5 (or theorem 2 and 6), it suffices to choose a finite $T$ such that the average optimization error is controlled within $O(\varepsilon)$ across all episodes $K$ in order to achieve the $\varepsilon$-suboptimality of the output mixture policy.
> > >
> > > - *Theorem 4: why there is constraint violations in the case of strict feasibility? (c.f. line 14 saying zero violation)*
> > >
> > > In the strict feasibility regime, while our algorithm is guaranteed to output an $\varepsilon$-suboptimal and zero-violation policy after $\tilde{O}(\frac{SAH^5}{\varepsilon^2\zeta^2})$ episodes (c.f. lines 144-146, equation 4, and theorem 6), the algorithm itself may incur constraint violation in some online learning episodes. We upper bound the algorithm's cumulative constraint violation with cancellation in theorem 4, and further upper bound it by 0 with chosen values of parameters in theorem 6, proving that the final output mixture policy indeed achieves zero violation. However, to ensure there is no constraint violation in all learning episodes, we require additional assumptions, such as assuming the knowledge of a safe policy $\pi^0$ and its underlying true cost value $V_{1,c}^{\pi^0}(s_1)$ [1,2], or assuming the knowledge of a safe action for all states [3].
> > >
> > > ---
> > >
> > > References
> > >
> > > [1] Tao Liu, Ruida Zhou, Dileep Kalathil, P. R. Kumar, and Chao Tian. Learning policies with zero or bounded constraint violation for constrained mdps. In Proceedings of the 35th International Conference on Neural Information Processing Systems, NIPS ’21, 2021.
> > >
> > > [2] Archana Bura, Aria Hasanzade Zonuzy, Dileep Kalathil, Srinivas Shakkottai, and Jean-Francois Chamberland. Dope: doubly optimistic and pessimistic exploration for safe reinforce- ment learning. In Proceedings of the 36th International Conference on Neural Information Processing Systems, NIPS ’22, 2022.
> > >
> > > [3] Sanae Amani, Christos Thrampoulidis, and Lin Yang. Safe reinforcement learning with linear function approximation. In International Conference on Machine Learning, pages 243–253. PMLR, 2021.

---

> > > > ### Comment · Reviewer_JXZe · 2025-08-04
> > > >
> > > > thanks. I have no further questions.

---

### Official Review · Reviewer_A1jm · 2025-07-02

**Clarity:** 3
**Significance:** 2
**Originality:** 2
**Rating:** 4
**Confidence:** 4

**Summary:**

The paper considers the online learning problem for episodic constrained MDPS.
For the proposed online learning algorithm, which does not require access to a generative model(as in [3]), they provide regret bounds, constraint violation bounds, and sample complexities for relaxed and strict constrained MDP scenarios.  The algorithm and proofs leverage many existing ideas from constrained MDPs and unconstrained MDPs to improve the regret bounds compared to other existing algorithms.

**Questions:**

**Technical Questions**

- The algorithm uses optimistic rewards and cost functions along with the empirical model, even though the reward and cost functions are known. In unconstrained RL (UCRL2 variants ) as well as in constrained RL [1], optimistic estimates are used for the unknown quantities ($P$ in the case). Can the author provide their intuition on why they do not use optimistic estimates of P instead of the MLE estimate, as well as use optimistic reward and cost when they are assumed to be known?  When optimism is required and when empirical estimate suffices? This is similar to [3] but more surprising as they assume access to a generative model.


- As mentioned earlier, even though the algorithm is shown to have improved pre-constants in regret, it does not give a complete picture of the comparison of various model-based constrained RL algorithms.  A common theme in many unconstrained, tabular RL papers is to provide empirical comparison with existing algorithms in terms of regret and computational complexity; this has been missing in most Tabular constrained MDP papers, including this one. Can authors potentially address this missing aspect in this line of papers, which can provide new insights into these algorithms?

- Why is $\pi^{\tau,*}$ a deterministic policy (Line 233)?
- As suggested in Lines 55-56 ("large-scale state spaces"), what aspect of the algorithm makes it more suitable for large state spaces compared to the previous algorithms?


**Writing / Presentation Questions (and Suggestions)**
- As found in many theoretical RL papers these days (e.g., see Table 1 in [1] and [2]), can the author provide a table comparing regret bounds, constraint violation, and sample complexity of various algorithms in the related work/ introduction section? Only mentioning that the existing algorithm provides sublinear regrets (lines 40-41) does not give a full picture when the primary contribution of the paper is improving pre-constants in regret bounds.
- Would it be better to use some other symbol instead of $T$ for the number of primal-dual updates? Since $T$ is usually used for time ($HK$) in regret bounds, it can be potentially confusing.
- The title is perhaps too similar to [3]. I recommend that the authors consider some modifications that highlight the contribution of this paper, particularly compared to [3].

- While most of the paper is well-written and easy to follow, can the authors explain the "profile" mentioned in lines 167-168?
- Missing values in Line 552 after "With probability at least"
















**Reference**

[1]  Tao Liu, Ruida Zhou, Dileep Kalathil, P. R. Kumar, and Chao Tian. Learning policies with zero
or bounded constraint violation for constrained MDPs. In Proceedings of the 35th International
 Conference on Neural Information Processing Systems, NIPS ’21, 2021.

[2] Gen Li, Laixi Shi, Yuxin Chen, Yuantao Gu, and Yuejie Chi. Breaking the sample complexity
 barrier to regret-optimal model-free reinforcement learning. In Proceedings of the 35th International Conference on Neural Information Processing Systems, NIPS ’21, Red Hook, NY,
 USA, 2021.

[3] Sharan Vaswani, Lin Yang, and Csaba Szepesvári. Near-optimal sample complexity bounds
for constrained MDPs. In Advances in Neural Information Processing Systems, 2022

**Ethical Concerns:**

["NO or VERY MINOR ethics concerns only"]

**Final Justification:**

The paper has received a positive response from all reviewers. Most reviewers highlight that there is room for improvement in presentation and clarity of the paper. The authors addressed my questions and concerns adequately. I retain the current positive rating of the paper, hoping all the changes discussed during the rebuttal are included in the camera-ready version.

**Limitations:**

Yes

**Quality:**

3

**Strengths And Weaknesses:**

**Strengths**

- The paper addresses many theoritical aspects of the constrained MDP learning problems by providing regret bounds, constraint violation bounds and sample complexity in both relaxed and strict feasibility cases.
- The paper removes the requirement of access to generative model assumed in [3].
- The presentation and writing in this paper is good and easy to follow, particularly the inclusion of Section 4 provides lot of insights into the existing work and the algorithm.


**Weaknesses**
- The paper mentions that constrained MDP is an important setup for RL problems with safety concerns. The paper's contribution is purely algorithmic and theoritical. It lacks any experimental / simulation studies, comparison with existing algorithms. As seen in unconstrained RL algorithms, particularly UCRL variants, algorithm with similar regret bounds have been shown to have different empirical performance, so improved regret bounds alone are not sufficient to make a strong case for a new algorithm.

- Please see the questions section for potential weaknesses and questions.

**References**

[1]  Tao Liu, Ruida Zhou, Dileep Kalathil, P. R. Kumar, and Chao Tian. Learning policies with zero
or bounded constraint violation for constrained mdps. In Proceedings of the 35th International
 Conference on Neural Information Processing Systems, NIPS ’21, 2021.

[2] Gen Li, Laixi Shi, Yuxin Chen, Yuantao Gu, and Yuejie Chi. Breaking the sample complexity
 barrier to regret-optimal model-free reinforcement learning. In Proceedings of the 35th International Conference on Neural Information Processing Systems, NIPS ’21, Red Hook, NY,
 USA, 2021.

[3] Sharan Vaswani, Lin Yang, and Csaba Szepesvári. Near-optimal sample complexity bounds
for constrained MDPs. In Advances in Neural Information Processing Systems, 2022

---

> ### Author Rebuttal · Authors · 2025-07-31
>
> We thank the reviewer for the constructive feedback, insightful comments, and positive assessment of our work.
>
> We agree that empirical evaluation of CMDP algorithms is valuable. Our primary contribution of this paper is theoretical and to close the gap between upper and lower bounds for online tabular CMDPs using novel analysis techniques. While comprehensive empirical study is beyond the scope of this paper, we plan to conduct numerical experiments in future work.
>
> We respond to the reviewer's questions below:
> - *Can the author provide their intuition on why they do not use optimistic estimates of $P$ instead of the MLE estimate, as well as use optimistic reward and cost when they are assumed to be known? When optimism is required and when empirical estimate suffices?*
>
> We thank the reviewer for this excellent question. In our method, the UCB bonus $b_h(s,a)$ is indeed used to encode optimism for the transition model $P$. As shown in the optimism analysis (c.f. lemma 15), the bonus term is designed to cover the statistical error between $\hat{P}$ and $P$, guaranteeing an optimistic value estimates for any fixed policies (c.f. line 571). Because the UCB bonus $b_h(s,a)$ is analogous to reward and cost functions, mapping $(s,a,h)$ tuples to scalars, we add the bonus directly to rewards (or subtract from costs) and denote the estimated reward (or cost) values by $\hat{V}_{h,\tilde{r}}(s)$ (or $\hat{V}\_{h,\underline{c}}(s)$) to streamline the presentation.
>
> - *Why is $\\pi^{\\tau,\*}$ a deterministic policy (line 233)?*
>
> We thank the reviewer for catching this. The policy $\pi^{\\tau,\*}$ is in fact fixed, but not necessarily deterministic. We will correct the wording in line 233 to reflect that it may be randomized. Meanwhile, the policy $\pi^{\tau,*}$ remains independent of the online learning process, and this typo does not affect the soundness of our theoretical results.
>
> - *As suggested in Lines 55-56 ("large-scale state spaces"), what aspect of the algorithm makes it more suitable for large state spaces compared to the previous algorithms?*
>
> Our algorithm reduces the sample complexity by $\sqrt{SH}$ compared to the best existing results [1], and in scenarios with large state space $S$ and episode length $H$, the advantage of our algorithms will be significant.
>
> - *Can the author provide a table comparing regret bounds, constraint violation, and sample complexity of various algorithms in the related work/ introduction section?*
>
> We agree that a table will help readers. In the camera-ready version, we will add a table that compares the regret, constraint violation, and sample complexity of existing online CMDP algorithms to ours.
>
> - *Would it be better to use some other symbol instead of $T$ for the number of primal-dual updates? Since $T$ is usually used for time ($HK$) in regret bounds, it can be potentially confusing.*
>
> We appreciate the reviewer for the kind advice. We will refine the notations in the camera-ready version to eliminate ambiguity.
>
> - *The title is perhaps too similar to [2]. I recommend that the authors consider some modifications that highlight the contribution of this paper, particularly compared to [2].*
>
> We thank the reviewer for the constructive comment. We will revise the title to highlight the contributions of our work.
>
> - *While most of the paper is well-written and easy to follow, can the authors explain the "profile" mentioned in lines 167-168?*
>
> A profile is a “snapshot” of the online learning process that records the visitation counts $N_h(s,a)$ for all $(s,a,h)$ tuples (c.f. line 168).
>
> - *Missing values in Line 552 after ``With probability at least'':*
>
> We thank the reviewer for catching this typo. We will add the probability $1-2\delta'$ to line 552.
>
> ---
>
> Reference
>
> [1] Archana Bura, Aria Hasanzade Zonuzy, Dileep Kalathil, Srinivas Shakkottai, and Jean-Francois Chamberland. Dope: doubly optimistic and pessimistic exploration for safe reinforcement learning. In Proceedings of the 36th International Conference on Neural Information Processing Systems, NIPS ’22, 2022.
>
> [2] Sharan Vaswani, Lin Yang, and Csaba Szepesvári. Near-optimal sample complexity bounds for constrained MDPs. In Advances in Neural Information Processing Systems, 2022.

---

> > ### Comment · Reviewer_A1jm · 2025-08-05
> > **Response to the Rebuttal**
> >
> > I thank the authors for providing requested clarifications in the rebuttal. I will update my review accordingly, while retaining the current score. I recommend the authors to incorporate all the modification discussed in the rebuttal.
> >
> > Reviewer A1jm

---

### Official Review · Reviewer_R9pq · 2025-07-03

**Clarity:** 2
**Significance:** 4
**Originality:** 3
**Rating:** 5
**Confidence:** 2

**Summary:**

The authors improve upon known results from constrained reinforcement learning. Specifically, by improving the traditional bonus-driven approach with a primal-dual intuition, the authors improve the sample complexity for learning policies in the no-violation as well as the small-violation regimes.

**Questions:**

Can your results be stated in another way that avoids dependency on the stater coefficient? Also, do the authors have any thoughts on the optimality of their results? Specifically, for the $\epsilon$-violating regime, would it be possible to achieve the same sample condition while achieving even smaller than $\epsilon$ violation?

**Ethical Concerns:**

["NO or VERY MINOR ethics concerns only"]

**Final Justification:**

I would agree with many of the other reviewers that the results in this work are very good, but the paper could use some significant revisions to improve clarity. Given the author's responses on matching lower bounds, I think these technical results are even better than I originally thought and certainly outweigh any writing issues, but would still recommend the authors make some edits for the camera-ready version.

**Limitations:**

Yes

**Quality:**

3

**Strengths And Weaknesses:**

The paper's main strength is the construction of an algorithm with improved sample complexity guarantees. In addition to the theoretical advancements, using primal-dual intuitions within the bonus-driven approach is a very nice idea. The main weakness of this paper is the presentation; the paper is very mathematically driven and terse. It could benefit from some improvements in writing for clarity. It could also be argued that the improvements in the sample complexity are not drastic, but I believe they are very meaningful nonetheless.

---

> ### Author Rebuttal · Authors · 2025-07-31
>
> We thank the reviewer for the constructive feedback, insightful comments, and positive assessment of our work.
>
> We will improve the exposition and expand the high-level intuition of our methods to enhance overall clarity. Although the improvement of $\sqrt{SH}$ in the sample complexity compared to the best existing bounds is sublinear in form, it closes a long-standing theoretical gap between the existing upper bounds for online CMDPs and the information-theoretic lower bound. Moreover, its benefits scale with the size of the state space and the episode length, yielding especially significant improvements in large‑state‑space regimes.
>
> We respond to the reviewer's questions below:
>
> - *Can your results be stated in another way that avoids dependency on the stater coefficient?*
>
> For the relaxed feasibility setting, the sample complexity does not depend on the Slater constant $\zeta$. For the strict feasibility setting, the sample complexity must depend on the Slater constant $\zeta$. It is required both by our upper bound analysis and the lower bound proved in [1]. Intuitively, in the strict feasibility setting, an estimation error of $\varepsilon$ in the cost domain can lead to a $\frac{H}{\zeta}\varepsilon$-suboptimality in the reward domain. Therefore, ensuring an $\varepsilon$-suboptimality in reward requires controlling estimation error of cost to within $\varepsilon' = \frac{\zeta}{H}\varepsilon$. Achieving that precision in MDPs costs $\Omega(\frac{SAH^3}{\varepsilon'^2}) = \Omega(\frac{SAH^5}{\varepsilon^2\zeta^2})$ samples, forcing the $\zeta$-dependence.
>
> - *Do the authors have any thoughts on the optimality of their results? Specifically, for the $\varepsilon$-violating regime, would it be possible to achieve the same sample condition while achieving even smaller than $\varepsilon$ violation?*
>
> Our upper bounds for sample complexity in both regimes match the corresponding lower bounds (up to logarithmic factors), so our results are minimax optimal (up to logarithmic factors). In the relaxed feasibility setting, the sample complexity depends on the smaller of the reward suboptimality and the cost violation tolerances.
>
> ---
>
> Reference
>
> [1] Sharan Vaswani, Lin Yang, and Csaba Szepesvári. Near-optimal sample complexity bounds for constrained MDPs. In Advances in Neural Information Processing Systems, 2022.

---

> > ### Comment · Reviewer_R9pq · 2025-08-09
> > **Rebuttal Response**
> >
> > Thank you for your response! I would agree with many of the other reviewers that the results in this work are very good, but the paper could use some significant revisions to improve clarity. Given the matching lower bounds, I think these technical results are even better than I originally thought and certainly outweigh any writing issues, but I would still recommend the authors make some edits for the camera-ready version.

---

> > > ### Author Response · Authors · 2025-08-09
> > >
> > > We thank the reviewer's positive evaluation of our work. As both reviewers A1jm and JXZe remarked that "the paper is generally well-written" and "the presentation and writing in this paper is good and easy to follow", we respectfully disagree that this paper needs significant revisions. Nonetheless, we agree with the reviewer that minor edits can enhance clarity, and we will improve the presentation in the camera-ready version.

---

### Official Review · Reviewer_f6iM · 2025-07-03

**Clarity:** 2
**Significance:** 4
**Originality:** 3
**Rating:** 5
**Confidence:** 3

**Summary:**

In this work, the authors tackle safety in RL through the framework of constrained MDPs (CMDPs). A CMDP is essentially a Markov decision process equipped with a cost function. Safety is encoded through a bound on the maximum cost the agent must incur while maximizing its return. Precisely, the authors develop a theoretical framework for tabular episodic MDPs, equipped with sample complexity for regret bounds and constraints violations. They consider two settings: strict feasibility, where the safety bound needs to be strictly satisfied, and relaxed feasibility, where the user fixes an acceptable error term violating the safety bound. Notably, the authors provide an online algorithm. This is in contrast to previous work that relied on access to a generative model (an assumption hardly feasible in practice). To achieve the guarantees, the technique is based on a primal-dual formulation. Interestingly, the provided sample complexity in the relaxed feasibility setting matches the lower bound for unconstrained MDPs, while in the strict setting, it matches the lower bound for constrained MDPs with access to a generative model.

**Questions:**

- Could you explain why ‘fixing a profile that keeps track of visitation counts for all $(s, a, h)$ tuples makes $\hat{V}$ independent of $\hat{P}$’ (line 168)?
- Could you also explain how the double-batch update allows to avoid the number of profiles to be exponential in $K$?
- Could you please motivate a bit more the introduction of the USB/Bernstein-style bonus $b^{k, \pi}_{h, g}$ that is added to the reward function and subtracted from the cost estimate? Intuitively, how does this encourage exploration? And how does this allow us to underestimate the cumulative cost?
- Could you comment and explain the equality of Equation 8?
- In Algorithm 1, how do you obtain/compute exactly $\hat{\pi}_{t}^{k}$? We assume here to retrieve the policy maximizing Equation 8 (through a careful discretization of $\lambda$). In practice, I guess one should use an LP formulation or policy iteration, right?

**Ethical Concerns:**

["NO or VERY MINOR ethics concerns only"]

**Final Justification:**

As originally stated in my review, I believe that the theoretical results presented in this paper are significant.
Note that I disagree with reviewer JXZe on the fact that the episodic and tabular MDPs are too restrictive.
Indeed, providing theoretical sample complexity guarantees in those settings is the cornerstone of further theory that can be developed in more advanced and complicated settings. Furthermore, the majority of sample complexity results are developed in tabular settings.
Since the authors answered thoroughly to my questions, I raised my score, under the condition that (1) they implement the remarks and their answers in the final version of the paper, and (2) they polish the clarity and presentation of their results.

**Limitations:**

yes

**Quality:**

3

**Strengths And Weaknesses:**

# Strengths
- The theoretical results seem rather significant: in the relaxed setting, learning in a CMDP is as easy as learning in a classic MDP. On the other hand, in the strict setting, learning in the online setting is as easy as with generative access.
- The results presented cover two crucial axes: regret bound and constraint violation, for the two settings discussed above.
- The authors did a good job in presenting the related work, clearly exposing the problem setup and discussing why previous methods failed to achieve near-optimal sample complexity (beginning of Section 4).

# Weaknesses
- The main concerns I have about this paper are the lack of clarity and problems of presentation. In particular, this makes the paper very difficult to follow, especially for a non-expert reader in RL sample complexity/regret analysis.
- The paper is very dense (in particular, starting from Section 4) and extremely notation-heavy. Of course, this is a theoretical paper, and that’s the kind of work where formalities are necessary. However, finding the right balance between clarity and formality is key; in the main text, clarity is perhaps even more important as formalities, and notations make the paper very difficult to follow. Beyond the high number of symbols to digest, some of them are even reused for different purposes. For instance, $b$ is used both for the safety bound and for the UCB-style bonus. Taken out of context, it is very confusing. Another example is in Equation 8, where the authors write in a single line that the primal update is equal to subtracting the dual variable multiplied by the optimistically biased cost estimated. First, this is entirely written in the equation through subscript of the value function. This is very difficult to parse. Furthermore, this is written via several layers of subscripts, which makes it even more difficult to read. Why not detail this equality in the main text? Second, the authors do not explain why this equality holds. It should be made clearer.
- Starting at page 5 (in the methodology section), the paper starts to become very difficult to follow due to the high number of notations, subscripts, superscripts, and the lack of motivation when introducing new concepts and algorithmic choices. Those are later on justified in the proofs, but to me, some of them should be motivated beforehand. To be clear, the design choice of the algorithms is rightfully _described_, but rarely motivated. As an example, the authors describe a double batch update scheme to learn an empirical transition matrix and briefly mention that this is to make $\hat{V}$ independent from $\hat{P}$. How? This is not justified. Another example is the reason why a mixture of policies is taken in Algorithm 1. This is not motivated before diving into the proofs.
- If I am not mistaken, the algorithm assumes knowledge of the reward and cost functions (they are given as input of Algorithm 1). Unless I am mistaken, this is not clearly stated in the main text. I think this also deserves a (at least brief) discussion, even if it is considered ‘standard’ in theoretical RL — in practice, that’s not.
- The paper ends abruptly with a Theorem and its proof. I would really have liked to have a final formal discussion about the results presented in Section 5 and their implications on the ongoing research.

---

> ### Author Rebuttal · Authors · 2025-07-31
>
> We thank the reviewer for the constructive feedback, insightful comments, and positive assessment of our work.
>
> We respond to the reviewer's concerns below:
>
> - *Clarity and presentation:*
>
> We appreciate the reviewer's feedback on clarity. We are confident that the clarity concerns can be fully addressed in the camera-ready version through streamlined exposition, more in-line intuition, and improved presentation, so the technical content remains rigorous and more accessible.
>
> - *Notation reuse:*
>
> We thank the reviewer for pointing this out. We will introduce distinct notations for the constraint constant and the UCB-style bonus to eliminate ambiguity.
>
> - *Regarding equation 8:*
>
> We agree that the equality should be spelled out in the main text. A full derivation will be added. Please refer to our responses to the reviewer's Question 4 below for detailed explanations.
>
> - *Motivation for the doubling batch update and mixture policy:*
>
> Please refer to our responses to Questions 1 and 2 for the motivation behind the doubling batch update. Regarding the mixture policy, since the policies obtained from each primal update are deterministic, and CMDPs often require stochastic policies to satisfy constraints exactly, we adopt mixture policies to introduce the necessary randomization.
>
> - *If I am not mistaken, the algorithm assumes knowledge of the reward and cost functions (they are given as input of Algorithm 1). Unless I am mistaken, this is not clearly stated in the main text. I think this also deserves a (at least brief) discussion, even if it is considered ‘standard’ in theoretical RL — in practice, that’s not.*
>
> We state the assumption of known reward and cost functions in Section 3 (line 118-120). Learning reward and cost functions in CMDPs does not affect the leading terms of the final sample complexity [1], and we introduce this assumption to keep the exposition concise and focus on the core contributions without loss of generality.
>
> - *The paper ends abruptly with a Theorem and its proof. I would really have liked to have a final formal discussion about the results presented in Section 5 and their implications on the ongoing research.*
>
> We agree that the paper would benefit from a more conclusive discussion, and we will include a brief summary of our results and implications for future research.
>
> ---
>
> We respond to the reviewer's questions below:
> 1. *Why fixing a profile makes $\hat{V}$ independent of $\hat{P}$:*
>
> Recall that a profile is a “snapshot” of the online learning process that records the visitation counts $N_h(s,a)$ for all $(s,a,h)$ tuples (c.f. line 168). In unconstrained MDPs, we compute: $\hat{V}\_h(s) = \max\_a \hat{Q}\_h(s,a) = \max\_a\\{r\_h(s,a) + b\_h(s,a) + \hat{P}\_{s,a,h}\hat{V}\_{h+1}\\}$. When fixing a profile, the value estimate $\hat{V}\_{h+1}$ can be deterministically computed from $\\{\hat{P}\_{h'}\\}\_{h'=h+1}^H$ and thus independent of $\hat{P}\_h$.
>
> 2. *How does the double-batch update avoid the number of profiles to be exponential in $K$:*
>
> The number of profiles equals the number of strictly increasing paths of visitation‑count updates, which is exponential in the number of updates. Adopting the doubling batch update reduces the number of updates from $K$ to $\log\_2 K$, thereby avoids profile numbers being exponential in $K$ (see Lemma 4.2 of [2] for a formal proof).
>
> 3. *How does the UCB-style bonus encourage exploration while underestimate cost:*
>
> The UCB-style bonus shrinks with more visitation counts, assigning higher value to less-visited $(s,a,h)$ tuples, and thus encouraging exploration. By subtracting UCB-style bonus from estimated cost values, we offset potential overestimation of $\hat{P}$ by statistical error, effectively leading to an underestimation of cumulative costs.
>
> 4. *Comment and explain the equality of Equation 8:*
>
> We provide the detailed explanation to equation 8 here, which will be added to the appendix in the camera-ready version:
>     Note that $\\hat{V}\_{1,g}^\\pi(s) = \\mathbb{E}\_{\hat{P},\pi}\left[\sum\_{t=1}^H g(S\_t, A\_t)|S\_1 = s\right]$ for any reward-like function $g$ mapping $(s,a,h)$ tuples to scalars, then we have
>        $\\hat{\\pi}\_t^k = \\arg\\max\_\\pi \\mathbb{E}\_{\\hat{P},\\pi}\left[\sum\_{t=1}^H \tilde{r}(S\_t, A\_t)|S\_1 = s\_1\right] - \\hat{\\lambda}\_t^k\cdot\\mathbb{E}\_{\hat{P},\pi}\left[\sum\_{t=1}^H \\underline{c}(S\_t, A\_t)|S\_1 = s\_1\right] + \hat{\lambda}\_t^k\cdot b'$ $= \\arg\\max\_\\pi \\mathbb{E}\_{\\hat{P},\pi}\left[\sum\_{t=1}^H (\tilde{r} - \hat{\lambda}\_t^k \underline{c})(S_t, A_t)|S_1 = s_1\right] = \\arg\\max\_\\pi \hat{V}\_{1,\tilde{r} - \hat{\lambda}\_t^k\underline{c}}^\\pi(s_1)$.
>
> 5. *How do you obtain exactly $\hat{\pi}\_t^k$:*
>
> In the primal update $\hat{\lambda}\_t^k$ is fixed, and we obtain $\hat{\pi}\_t^k$ from the unconstrained MDP with Lagrange-augmented rewards (c.f. equation 8). The policy can be exactly solved by backward inductions as in standard unconstrained MDPs [2].
>
> ---
>
> Reference
>
> [1] Sharan Vaswani, Lin Yang, and Csaba Szepesvári. Near-optimal sample complexity bounds for constrained MDPs. In Advances in Neural Information Processing Systems, 2022.
>
> [2] Zihan Zhang, Yuxin Chen, Jason D Lee, and Simon S Du. Settling the sample complexity of online reinforcement learning. In Proceedings of Thirty Seventh Conference on Learning Theory, Proceedings of Machine Learning Research. PMLR, 2024.

---

> > ### Comment · Reviewer_f6iM · 2025-08-05
> >
> > Thank you for answering my questions.
> > I have no further questions.

---

### Note · Authors · 2025-08-16

We thank the reviewers for their thoughtful and constructive feedback and positive assessment of our work, and we thank the Area Chairs for effectively coordinating the reviewing process.

Our work advances the theory of safe RL for tabular online CMDPs by closing the gap between existing upper bounds and known lower bounds in both regimes we study. Methodologically, our analysis integrates a primal–dual framework with a doubling batch update scheme that decouples estimation correlations, and the use of mixture policies to satisfy constraints. We believe both our algorithm designs and analysis techniques will inform future safe RL algorithm design and scale favorably with state space and horizon.

The reviewers unanimously affirmed the soundness and significance of our results, and the remaining comments are editorial refinements to further improve clarity. We will incorporate all presentation suggestions raised in the reviews and discussions in the camera-ready version, such as cleaning up a few overloaded notations, fixing minor typos, and adding a brief concluding discussion to highlight the implications.

We are grateful for the community's time and effort and will reflect these improvements in the final camera-ready version.

---

### Decision · Program_Chairs · 2025-09-17

**Decision:**

Accept (poster)

**Comment:**

The paper considers the problem of learning under constrained MDP where constraints can be seen as enforcing safety for instance. The paper is theoretically strong and reviewers agree with this. Reviewers do mention that the paper might be dense to read and it would be good for the authors to fix before final version. That said I believe the paper is clearly above the bar for publication at Neurips.